



# Investigation of the effects of the Greek extreme wildfires of August 2021 on air quality and spectral solar irradiance

Akriti Masoom[1], Ilias Fountoulakis[2, 3, 4], Stelios Kazadzis [1], Ioannis-Panagiotis Raptis[5],
Anna Kampouri[2, 6], Basil Psiloglou[5], Dimitra Kouklaki[4], Kyriakoula Papachristopoulou[2, 4],
Eleni Marinou[2], Stavros Solomos[3], Anna Gialitaki[7,8], Dimitra Founda[5], Vasileios Salamalikis[9],
Dimitris Kaskaoutis[5], Natalia Kouremeti[1], Nikos Mihalopoulos[5], Vasilios Amiridis[2],
Andreas Kazantzidis[9], Christos Zerefos[3, 10, 11, 12], and Kostas Eleftheratos[4, 10]

[1]Physikalisch-Meteorologisches Observatorium Davos / World Radiation Center (PMOD/WRC), Dorfstrasse, 7260 Davos Dorf, Switzerland
[2]Institute for Astronomy, Astrophysics, Space Applications and Remote Sensing, National Observatory of Athens, Athens, GR-15236, Greece
[3]Research Centre for Atmospheric Physics and Climatology, Academy of Athens, Athens, Greece
[4]Department of Geology and Geoenvironment, National and Kapodistrian University of Athens, Athens, GR-15784, Greece
[5]Institute for Environmental Research & Sustainable Development, National Observatory of Athens, I. Metaxa & Vas. Pavlou, P. Penteli, GR-15236 Athens, Greece
[6]Department of Meteorology and Climatology, Aristotle University of Thessaloniki, Thessaloniki, Greece
[7]School of Physics and Astronomy, Earth Observation Science Group, University of Leicester, Leicester, United Kingdom
[8]Laboratory of Atmospheric Physics, Physics Department, Aristotle University of Thessaloniki, Thessaloniki, Greece
[9]Laboratory of Atmospheric Physics, Department of Physics, University of Patras, GR 26500, Patras, Greece
[10]Biomedical Research Foundation of the Academy of Athens, GR-11527, Athens, Greece
[11]Mariolopoulos-Kanaginis Foundation for the Environmental Sciences, GR-10675, Athens, Greece
[12]Navarino Environmental Observatory (N.E.O.), Costa Navarino, GR-24001, Messinia, Greece

**Correspondence:** Kostas Eleftheratos (kelef@geol.uoa.gr)

**Abstract.** In August 2021, a historic heatwave was recorded in Greece which resulted in extreme wildfire events that strongly affected the air quality over the city of Athens. Saharan dust was also transferred over Greece in the same period due to the prevailing southern winds. The impact of these events on air quality and surface solar radiation are investigated in this study. Event characterization based on active and passive remote sensing instrumentation has been performed. The study shows that significantly increased levels of air pollution were recorded during the end of July/first week of August. The smoke led to unusually high AOD values (up to 3.6), high Ångström Exponent (AE) (up to 2.4) and a strong and negative dependence of single scattering albedo (SSA) on wavelength that was observed to decrease from 0.93 at 440 nm to 0.86 at 1020 nm signifying the presence of strong absorbing aerosols. While, the dust event led to high AOD (up to 1.4), low AE (up to 0.9) and positive dependence of SSA on wavelength that was observed to increase from 0.89 at 440 nm to 0.95 at 1020 nm indicating large forward scattering due to coarse particles. Furthermore, the analysis of the smoke aerosol optical properties during the transfer from the source to a distance of about 240 km revealed that the SSA and AE changed significantly during the transfer, which lasted approximately 9 h. The transport of the plume led to an impressive change in the spectral shape of SSA whose value significantly increased pointing to the aging of smoke and the dilution of plumes while the transport. The impact of dust and



smoke on spectral solar irradiance reveals significant differences in the spectral shape of attenuation caused by the two different aerosol species. The attenuation of solar irradiance in UV-B irradiance was found to be least in case of dust and highest due to smoke (up to 60% or more) and intermediate in the case of a mixture of smoke and dust. The attenuation was comparatively less in NIR region (mostly within 20% but it even reached up to 40% in the presence of smoke) and VIS region (but greater than NIR region). Also, the AOD variations from climatology led to decrease in UV Index up to 53%, in vitamin-D up to 50%, in photosynthetically active radiation up to 21% and in GHI up to 17%, with implications on health, agriculture and energy. This study highlights the wider impacts of wildfires that are part of the wider problem of the Mediterranean countries, whose frequency is predicted to increase in view of the projected increasing occurrence of summer heatwaves.

## 1 Introduction

Climate change is becoming a harsh reality and leading to climate havocs one of which is the increased frequency of occurrence of large–scale wildfires around the globe which affect both environment and human life (Weilnhammer et al., 2021). Wildfires lead to loss of land vegetation, worsened air quality and affects the ecosystems, societies, economies and climate (Jaffe et al., 2013; Jolly et al., 2015) and there has been concerns about the frequency of occurrence of such events in the recent past (Ganor et al., 2010; Forzieri et al., 2017). Extreme weather events of severe heat waves (Perkins-Kirkpatrick and Lewis, 2020; Fischer et al., 2021), which are more prominent in the Southern and southeastern Europe (Giorgi and Lionello, 2008; Fernandez et al.; Forzieri et al., 2017; Füssel et al., 2017; Weilnhammer et al., 2021), act as fuel for other extreme events like wildfires. The probable causes of ignition of wildfires can be categorized into lightning–induced and human-caused.

A wildfire event leads to a sudden rise in harmful constituents into the atmosphere consisting of particulate matter and gaseous pollutants such as nitrogen oxides, carbon monoxide, greenhouse gases and volatile organic compounds (Andreae and Merlet, 2001; Knorr et al., 2017; Fernandes et al., 2022). Among these, greenhouse gases, having longer lifetime, impact the global climate, while aerosols, having short lifespan, has regional effects. Some of these atmospheric pollutants get transported to surrounding areas far from the source. The long–range transport of the wildfire smoke can lead to a change in the chemical composition of the plume and also affect the local and regional air quality upon the planetary boundary layer entrainment (Colarco et al., 2004; Wu et al., 2021). Post wildfire, the air quality pose serious health issues due to the high gaseous and particulate pollutant levels that imposes serious threats of asthma, respiratory diseases, cardiovascular effects, and lung cancer via human inhalation exposure (Manisalidis et al., 2020; Rice et al., 2021). Hence, a robust and a more coherent understanding of consequences of such events is crucial. Biomass burning also reduces the amount of solar radiation as presented in the study by Rosário et al. (2022) that showed the mean drop in solar radiation was up to $200\,\mathrm{W\,m^{-2}}$. In another study by Park et al. (2018), it was found that smoke reduces the UV actinic flux and the spectral response also depends upon the type of smoke. Moreover, Arola et al. (2007) found that biomass burning aerosols led to about 35 % diminishing of surface UV irradiance while the reduction was comparatively smaller for total solar radiation.

The Mediterranean is considered a "climate change hotspot" (Founda et al., 2022; Zittis et al., 2022) due to it's faster warming rates, as compared to the global average; as well as an increase in the frequency of heat waves followed by forest fires



and prolonged droughts. Mediterranean region is also susceptible to increased aridity as a result of climate change (IPCC2022; Turco et al., 2018; Guiot and Cramer, 2016). Occurrence of fires leads to water stress that in turn reduces the post–fire vegetation recovery (Puig-Gironès et al., 2017; Cruz and Moreno, 2001; Pratt et al., 2014; Vilagrosa et al., 2014; Pausas et al., 2016)

leading to an expansion of shrublands as a combined effect of fire and drought (Batllori et al., 2017, 2019; Baudena et al., 2020). Different aerosol optical properties and mixtures of them have been investigated in various papers using various ways some of which are discussed here. In Castagna et al. (2021), the authors analysed the 2017 summer wildfires in the Calabria Region (South Italy), which resulted in the largest burned area of the last decade (2008-–2019), estimated to be more than 1679 hectares of forests and shrub land. The impact of the wildfires on the air quality, ecosystem and human health was analysed

using the carbon monoxide and black carbon measurements at the high–altitude station of Monte Curcio. In Gómez-Amo et al. (2017), the authors studied two wildfires in Spain that occurred near Valencia during 29—30 June affecting 48.500 hectares of land.

The summer season in the Mediterranean region witnesses frequent dust activities and the dust particles are favoured by stable weather conditions due to the absence of precipitation and depressions (Nastos, 2012). The eastern Mediterranean region

has a marked seasonal cycle of the occurence of the Saharan dust with a maximum transport in spring and summer (Moulin et al., 1998; Rodríguez et al., 2001; Fotiadi et al., 2006; Meloni et al., 2007). In Papayannis et al. (2009), the authors presented a statistical analysis of Saharan dust for a 3–year period between 2004 and 2006 over Athens, Greece and they found that the Saharan dust related aerosol layers were prevalent for 79 days. In Marinou et al. (2017), the authors presented a statistical analysis of the 3D transport of Saharan dust towards Europe based on a 9–year dataset from CALIPSO (Cloud-Aerosol Lidar

and Infrared Pathfinder Satellite Observation) satellite. They show that Saharan dust layers arrive above Greece in altitudes between 2–6 km in spring (mean dust extinction coefficient values $\sim 70$ Mm$^{-1}$), between 3–6 km in summer ($\sim 50$ Mm$^{-1}$), and between 2–5 km in autumn ($\sim 40$ Mm$^{-1}$). Saharan dust effects in various sectors including health, aviation and solar energy have been presented in Monteiro et al. (2022) and references therein. Especially, studies estimating extreme dust events can result in Global Horizontal Irradiance (GHI) attenuation by as much as 40–50 % and a much stronger Direct Normal

Irradiance (DNI) decrease (80–90 %), while spectrally this attenuation is distributed to 37 % in the UV region, 33 % in the visible and around 30 % in the infrared (Kosmopoulos et al., 2017). Also, Papachristopoulou et al. (2022) showed that for the Eastern Mediterranean the average attenuation of dust in GHI/DNI using a 15–year climatology is $\sim$3 %/10 %.

The wildfires of summer 2021 in Greece were the most severe in a decade signifying a conflagration period of about 20 days in August, and were triggered by severe and prolonged heat waves, as discussed in a few recent studies. The study by Founda

et al. (2022) showed that the heat wave of 2021 was intense and persistent with the highest observed nighttime temperatures and cumulative heat which were also intensified due to urban heat island effect in Athens. A study of the Varympompi wildfire of 2021 in the northern suburbs of Athens by Giannaros et al. (2022) showed that it was characterized by unusual spread of fire followed by massive spotting as well as pyroconvection influence. This study analysed the physical drivers associated with this event using fire—atmosphere modeling system coupled with WRF–Fire and the relative contributions of weather, topography

and fuels. The development of pyroconvection and ignition was supported by dry and hot conditions that began emerging in late June that deteriorated further reaching the peak in July and resulted into the wildfires. The meteorological conditions



also supported the event including lack of significant precipitation and higher than average temperatures. Another study by Papavasileiou and Giannaros (2022) analyzed the pyroconvection using satellite data and found that there was a presence of pyrocumulus and pyrocumulonimbus for many hours.

The increase in the frequency of occurrence of these extreme wildfire events enhances the necessity of a more in-depth understanding of these phenomena and their impact on various domains. The analysis presented in this study focuses on such wildfire events that were prevalent throughout August of 2021 around the city of Athens. The study aspires to better analyze the wildfire smoke and simultaneous dust activity from in situ, remote sensing as well as modelling data and to analyze their respective and combined impact on spectral solar irradiance. The datasets used were collected from Athens, during the

Atmospheric parameters affecting SPectral solar IRradiance and solar Energy measurement campaign (ASPIRE), and from the PANhellenic GEophysical observatory of Antikythera (PANGEA) of the National Observatory of Athens (NOA). The wildfire events were investigated using active and passive remote sensing instruments, showing the complexity in different aerosol mixtures in Athens and the transport of the smoke to the PANGEA with a possible change in the chemical composition during the transport. Investigation of the effects of the fires on the solar spectral irradiance was performed using broadband and

spectral ground-based solar irradiance measurements and radiative transfer modelling.

    The aim of this work is to analyse the spatial and temporal aerosol spectral optical properties during the August 2021 wildfires in Athens and their effects on surface solar radiation. More specifically, the main objectives are: (1) to discuss the effect of the dust and smoke events on air quality, (2) to show how observations from different sensors can be combined to identify and study such events, (3) to study aerosol optical and microphysical properties during the events, (4) to investigate

changes in the composition of aerosols during their transport from Athens to the PANGEA observatory in Antikythera, and (5) to analyze the contribution of dust and biomass burning aerosols to the attenuation of spectral surface solar radiation over Athens. This paper is organized in four sections. Section 2 deals with the observational data and the methodology, followed by Sect. 3 that present the results and discussions and finally, Sect. 4 summarizes the findings from this study.

## 2   Data and Methodology

For a better understanding of the August 2021 wildfires in Greece, ground-based measurements, satellite images and radiative transfer modelling are used synergistically. This section deals with the description of the datasets used in this work as well as the methodology followed to study the wildfire event.

### 2.1   Ground–based measurements

In Athens, measurements that were collected during the intensive ASPIRE campaign have been used for the study. In addition

to the instruments that are permanently installed and operating at NOA's actinometric station (ASNOA) (located in the green area of Thissio in the center of Athens; 38.00° N, 23.73° E, 110 m above mean sea level), new instruments were installed in the context of the ASPIRE campaign. Ground-based remote sensing measurements are also performed at the Biomedical Research Foundation of the Academy of Athens (BRFAA), Greece (37.99° N, 23.78° E, at approximately 180 m a.s.l.), located in a



green area 4 km from the ASNOA. In situ air quality measurements are available from the stations of the Greek National Air
Pollution Monitoring Network. From PANGEA observatory (35.86° N, 23.31° E, 189 m a.s.l.) measurements collected with
the NOA's aerosol remote sensing facility have been used. Details on the equipment and measuring sites are provided in Table
1.

**Table 1.** Description of ground based measurements

| Quantity | Instrument/Network | Location | Description | Type |
|---|---|---|---|---|
| PM10, PM2.5 | GNAPMN | GAA | Daily | - |
| NO, $NO_2$ | GNAPMN | GAA | Hourly | - |
| BC, Scattering and Absorption Coeff | Aethalometer, Nephelometer | ASNOA | Daily | In situ aerosol |
| Columnar $NO_2$ | Pandora | ASNOA | | Spectral radiometer |
| Columnar $SO_2$ | Brewer | BRFAA | | Spectrophotometer |
| AOD, AE | Cimel | NOA, PANGEA | 15–min | Sunphotometer |
| SSA, Fine/Coarse AOD | Cimel | NOA, PANGEA | - | Skyphotometer |
| VSD | Cimel | NOA | - | Skyphotometer |
| Backscatter coefficient | Ceilometer | ASNOA | 910 nm | Vaisala |
| Backscatter coefficient | Lidar | PANGEA | 1064 nm | Polly-XT |
| Spectral irradiance | PSR | ASNOA | 300 -1020 nm | Spectral radiometer |
| UV-B irradiance | Brewer | BRFAA | 290-319 nm | Spectrophotometer |
| GHI, DHI | Pyranometer | ASNOA | 285 - 2800 nm | Thermopile |
| NIR irradiance | Pyranometer, PSR | ASNOA | PSR (290-700 nm) | Thermopile |
| Erythemal irradiance | Brewer, PSR | ASNOA | UV spectra up to 400 nm | Thermopile |
| Viatmin D dose | Brewer, PSR | ASNOA | UV spectra up to 330 nm | Thermopile |

### 2.1.1 Air quality

We have analysed air quality data for the Greater Athens Area (GAA) from the Greek National Air Pollution Monitoring
Network (GNAPMN). More specifically, we analyzed daily averages of particulate matter concentrations (PM10, PM2.5), as
well as hourly concentrations of nitrogen oxides (NO, $NO_2$) for the period July–August 2021 at eleven sites. Since data of NO
and $NO_2$ are provided on an hourly basis, and not on a daily basis as the PM data, we calculated daily mean concentrations of
NO and $NO_2$ when at least 12 hourly measurements were available. Analytical information on the stations contributing data to
the GNAPMN is provided by Grivas et al. (2008).Pandora instrument was used to retrieve columnar $NO_2$ (Herman et al., 2009).
Measurements from an MKIV, single monochromator Brewer spectrophotometer (Brewer#001) (Kerr, 2010; Kerr et al., 1985)
are used in this study which is operating on the roof of the BRFAA since July 2003. Brewer#001 measures automatically the
direct and the diffuse global irradiances, as well as the zenith–sky radiance in the ultraviolet (UV) and visible (VIS) spectral





regions (Eleftheratos et al., 2021; Diémoz et al., 2016). Total column $SO_2$ was retrieved from the Brewer instrument. In addition, ground–based measurements for spectral scattering and absorption coefficients were taken at the Air Monitoring

Station at Thissio by means of integrated nephelometer (TSI 3564) and aethalometer (AE–33) instruments. Nephelometer measures the spectral scattering coefficient (bsca,λ) at three wavelengths (450, 550 and 700 nm). Aerosol absorption was computed via AE–33 measurements at seven wavelengths (370, 470, 520, 590, 660, 880 and 950 nm), while the instrument also provides the BC concentrations and through the "aethalometer model", the fractions of BC related to biomass (or wood) burning (BCwb) and fossil–fuel compustion (BCff) (Liakakou et al., 2020). Quality controlled aerosol scattering, absorption,

BC and SSA values at Thissio are available on hourly basis (Kaskaoutis et al., 2021), while daily–averaged values are used in this study (1–20 August 2021).

### 2.1.2 Aerosol properties

ATHENS–NOA AERONET (Aerosol Robotic Network) station was operating from 2008 to 2021, with a CE318 sun/sky–photometer from Cimel Electronique (CIMEL#440) in operation during the study period. The columnar AOD, AE, fine/coarse

AOD, Single Scattering Albedo (SSA) and Volume Size Distribution (VSD) (Dubovik and King, 2000; Dubovik et al., 2006; Sinyuk et al., 2007), retrieved from AERONET Version 3 algorithm are used here (Giles et al., 2019). Level 1.0 direct sun products also were used in this study, since the automatic cloud screening algorithm for level 1.5 filtered out data related with the wildfire plumes, due to the very high temporal variations of the AOD. Manual control of sky images from the cloud camera confirmed that there were no clouds present. Accordingly, level 1.5 inversion products were used, since the strict criteria for

level 2.0 filters out a lot of useful retrievals in summer months, as explained thoroughly in Kazadzis et al. (2016). The approach of using lower–level data, theoretically increases the uncertainty of the retrievals, but the evidence provided by the collocated data of other sources provides a higher degree of assurance. Also, the climatological values of the aforementioned properties reported in previous studies (Raptis et al., 2020) are used as reference. The similar parameters (AOD, AE, Fine/Coarse AOD and SSA) were also collected for PANGEA observatory located in the remote island of Antikythera (35.86° N, 23.31° E, 189

m a.s.l.).

### 2.1.3 Clouds

The Q24M Mobotix (MOBOTIX) All-Sky Imager (ASI) was installed at ASNOA for observing the atmospheric conditions in Athens in the context of the ASPIRE campaign which operated from December 2020 to September 2022 having a temporal resolution of 10 s. Such kind of ASIs can be employed for performing cloud detection and characterization (Kazantzidis et al.,

2012; Wendt et al., 2022) and/or retrieving aerosol properties (Cazorla et al., 2009; Román et al., 2022; Kazantzidis et al., 2017). For the latter, (Kazantzidis et al., 2017) proposed a methodology for producing AOD at 440, 500 and 675 nm using RGB channels, the sun saturation area (a feature extracted from ASI images representative of AOD magnitude) and solar zenith position as inputs in a machine learning algorithm. This procedure was validated in the semi-arid areas of Almeria, Spain, showing promising results. In this study, the ASI images are used to separate clouds from wildfire smoke, augmenting

the AERONET datasets with cases erroneously characterized as clouds by the automated cloud–screening approach.



A Vaisala CL31 ceilometer is also installed at ASNOA which detects clouds from the attenuated backscatter profile (Kotthaus et al., 2016) and is part of the EUMETNET's program, "E–Profile" (ALCProfile). At PANGEA, the PollyXT–NOA lidar (part of EARLINET (European Lidar Network); (EARLINET)) and PollyNET (Raman and polarization lidar network); (POLLYNET)) are installed. The PollyXT–NOA lidar (Engelmann et al., 2016; Baars et al., 2016) is a multi–wavelength Raman–polarization

system with 24/7 operational capabilities, which provide vertical distributions of the particle backscatter coefficient at 355, 532, and 1064 nm, the extinction coefficient at 355 and 532 nm and the particle depolarization ratio at 355 and 532 nm, in altitudes from 0.2 up to 15 km above the surface. With these observations, and using well known methodologies, we can separate between aerosols or clouds, spherical and non–spherical particles in mixed aerosol layers (Tesche et al., 2009; Marinou et al., 2019), and between absorbing and non–absorbing aerosols, towards aerosol characterization and aerosol/cloud separation

(Baars et al., 2017). Using the aforementioned parameters, we identify the times and altitudes where cloud–free smoke layers are observed above the PANGEA observatory, and we use these measurements as a complimentary dataset in this study.

### 2.1.4 Solar irradiance

The Precision SpectroRadiometer (PSR), No. 007 operating at ASNOA since 2016 is a high precision and accuracy state–of–the–art spectrometer (details are provided in the Appendix) is used to retrieve spectral irradiance used in this study. It measures

irradiance in the spectral range 300—1020 nm with an average step of 0.7 nm and spectral resolution in the range of 1.5—6 nm (depending on the measured wavelength) (Raptis et al., 2018; Gröbner and Kouremeti, 2019). The uncertainty budget of the instrument is less than 1 % in VIS, less than 1.7 % in UV–A and higher than 2 % in UV–B (Gröbner and Kouremeti, 2019). UV–A (315—400 nm) and the total UV radiation (290—400 nm) and the Photosynthetically Active Radiation (PAR) (400–700 nm, has been calculated from PSR measurements.

The Brewer, whose general description has been provided in Section 2.1.1, was used to retrieve the spectral UV–B measurement of the global solar irradiance which is performed with a frequency of about half an hour. The uncertainty in the Brewer measurements is estimated to 5 % for wavelengths above 305 nm and solar zenith angles lower than 70° (Garane et al., 2006). UV–B was obtained from the Brewer as the integral of the spectral measurements in the range 290–315 nm. Two pyranometers used in this study are of the type Eppley PSPs (Precision Spectral Pyranometers, S/Ns: 26069, 26070) that perform continuous

measurements of the broadband global and diffuse horizontal irradiances (GHI, DHI) in the spectral range 285–2800 nm, at ASNOA since 1986 (details are provided in Appendix). The maximum daily error (daily integral) expected from these thermopile pyranometers is about 1–2 % (Hulstrom, 2003). These instruments have also imperfect angular response (Gueymard and Vignola, 1998) and hence, a model–based correction for this effect was applied using a methodology similar to Bais et al. (1998).

Near Infrared irradiance (700–3000 nm) was calculated from the difference between the GHI measurements from the pyranometer and the calculated integral of the PSR measurements in 290–700 nm spectral range. The erythemal irradiance was calculated as the product of the UV spectra measured by the Brewer and the PSR with the action spectrum proposed by the International Commission of Illumination (CIE) (McKinlay and Diffey, 1987; Webb et al., 2011). The effective dose for the production of pre–vitamin D3 in the human skin (hereon referred as vitamin D dose) was calculated similarly to the erythemal



irradiance but using the respective effective spectrum both using a spectral extension correction technique proposed by Fioletov et al. (2003).

## 2.2    Satellite and reanalysis data

### 2.2.1    Copernicus Atmospheric Monitoring Service (CAMS)

The Copernicus Atmospheric Monitoring Service (CAMS) reanalysis product (Inness et al., 2019) are used to identify the
dominant aerosol type during August 2021 over Athens. Total aerosol optical depth, dust aerosol optical depth and organic matter aerosol optical depth at 550 nm were collected and analyzed for a 2x2 pixel area centered over Athens for a month period in August 2021. The CAMS data are available at an interval of 3-h on a regular lon/lat grid (0.75° x 0.75°) and is retrieved using the CDS API service Copernicus Atmosphere Data Store (ADS).

### 2.2.2    Meteosat Second Generation (MSG)

The SEVIRI instrument onboard geostationary MSG (Meteosat Second Generation) satellites of EUMETSAT provides full earth disc data at different channels every 15 min. In this analysis, the European HRV cloud RGB product was utilized, which is a product based on the High Resolution Visible and IR10.8 SEVIRI channels. This data is advantageous for cloud monitoring in high resolution. These images were analysed for August 2021 in order to identify the events, the initiation of the wildfires and the plume transport.

**2.2.3    MERRA2**

For the identification of the dust transfer over Athens and Antikythera, the total dust optical thickness at 550 nm from Modern-Era Retrospective analysis for Research and applications version 2 (MERRA-2) has been used (GMAO). The specific reanalysis product is available on a global scale with a temporal resolution of 1 hour and at a grid resolution of 0.5° × 0.625° (latitude × longitude). The data used in this analysis includes specific days in August 2021 for latitudes 25° N – 50° N and
longitudes 10° W – 40° E which was obtained from the Giovanni platform maintained by National Aeronautics and Space Administration (NASA) (GSFS).

## 2.3    Modeling

### 2.3.1    Spectral surface solar radiation

The disort pseudospherical approximation (Buras et al., 2011) of the UVSPEC radiative transfer model that is included in the
libRadtran v2.4 package (Emde et al., 2016) was used to simulate the spectral solar irradiance in the range 290 – 3000 nm. Radiative transfer simulations were performed for August for the coordinates of the actinometric station of Thissio with a temporal resolution of 15 min. The Libradtran simulations were performed for three different groups of inputs.



Case (a): In the first case, the simulations were performed for the UV region with SSA=0.85, and for the VIS and NIR regions with SSA=0.95. The inputs including AOD (at 340 nm for UV and at 500 nm for wavelengths above 400 nm), AE (440 – 675 nm), and total column of WV were obtained from the CIMEL. The CIMEL measurements were interpolated to the time of the simulations (i.e., for entire August with a step of 15 mins).

Case (b): In the second case, climatological values of AOD (at 340 nm for UV and at 500 nm for wavelengths above 400 nm), AE (440 – 675 nm), SSA (average of SSA at 440 nm and 675 nm) and total column of WV were used which were derived by analyzing CIMEL measurements for 2008 – 2018 (Raptis et al., 2020).

Case (c): The third case uses AOD=0 and total column of WV from CIMEL as inputs. The total column of ozone (TCO) from the Brewer#001 was also interpolated to time of the simulations and was used as input in all three cases. A default concentration of 420 ppb was assumed for $CO_2$. Surface albedos used were 0.05 and 0.2 in UV and VIS, respectively.

In all cases simulations were performed for cloudless conditions assuming climatological profiles of atmospheric molecules corresponding to mid-latitude summer (Anderson et al.) and climatological profiles of the aerosol optical properties (Shettle, 1990). The extraterrestrial spectrum proposed by Kurucz (1994) was used for the simulations.

### 2.3.2 Aerosol source and transport

For analyzing the source and transport of the wildfire plumes, HYSPLIT and FLEXPART models were used. The HYSPLIT model uses a hybrid of Lagrangian and Eulerian approaches. HYSPLIT is used over regional to global scale to account for the transport of pollutants, their dispersion and deposition. In this analysis, 72 h back–trajectories ending at 12 UTC for Athens and PANGEA were generated using GDAS meteorological data at 7 levels varying from 500 m to 3500 m with an interval of 500 m above ground level. Also, the Lagrangian particle dispersion model FLEXPART-WRF (Brioude et al., 2013) was run in a backward mode for a 72 hours period. A total of 10,000 particles released at 9 altitudes (between 500 m to 4.5 km above ground level), at the PANGEA ground station. FLEXPART has been used in a large number of similar studies on long-range atmospheric transport (Stohl et al., 2005; Solomos et al., 2015, 2019; Kampouri et al., 2021). The FLEXPART simulations were driven by hourly meteorological fields from the Advanced Research WRF (ARW) model version 4 (Skamarock et al.). The WRF–ARW spatial set up was at $20 \times 20$ km horizontal grid spacing with $351 \times 252$ grid points, 31 vertical levels. Initial and boundary fields are from the National Centers for Environmental Prediction (NCEP) final analysis dataset (FNL) at $1° \times 1°$ resolution. Daily updated Sea Surface Temperature (SST) is taken from the NCEP $0.5° \times 0.5°$ analysis.

## 3 Results

### 3.1 Description of the Event

A series of wildfires that severely affected Athens occurred at three locations, namely Varympompi, North Evia and Villia. The three major wildfires (i.e., smoke sources) around Athens are shown in Fig. 1. The first source was in the north of Athens about 15 km near Varympompi and affected the air quality from August 4 to August 9 with about 8370 hectares of area burnt. The



second source was at a distance of about 50 km at the north of Evia Island and led to the worsening of air quality from August
255    3 to August 11 and with a burnt area of about 51000 hectares. Another source that affected the air quality from August 17 to
August 19 was at the Northeast at a distance of ∼20 km near Vilia with a resulting burnt area of 9400 hectares. In Giannaros
et al. (2022), the authors reported that a total area of 94,000 hectares was burnt collectively by five wildfires in 2021 in Greece.

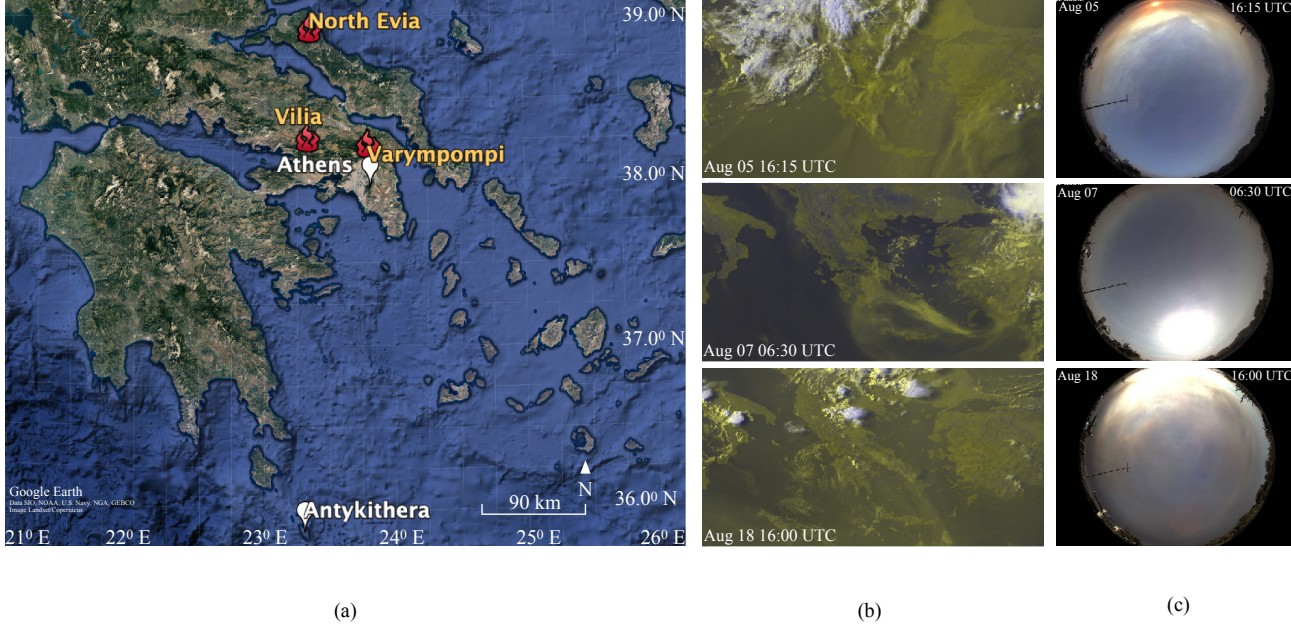

(a)                                              (b)                              (c)

**Figure 1.** (a) Map for the wildfire source sites (in red) and the study region (in white). Base map credits: ©2022 Google Earth. Identification
of the event on August 5, 7 and 18 using (b) MSG and (c) sky-camera images

According to NOA records, the summer (June – August) average temperature in Athens over the climatic period 1991–2020
was found to be around 28.5° C and the average daily maximum temperature about 34° C over the same period (Fig. A1a in
260    the Appendix). However, the period from the end of July to the beginning of August was marked by a very high temperature
surge, with positive air temperature anomalies of the order of 10° C compared to the long-term average (34° C) and even
reaching up to 44° C. These results are in agreement with the results reported by (Founda et al., 2022). Moreover, the relative
humidity from the end of July to early August was observed to be well below its climatic value (summer average humidity from
1991 to 2020, Fig. A1 b). Apart from temperature and relative humidity, the maximum wind speed during the end of July to
early August was found to be around 5.4 m/s, well below 5 Beaufort (8.0–10.7 m s$^{-1}$) (Fig. A1c and d in appendix). Yet, total
precipitation in Athens from March to July 2021 was found to be about 75 % lower than its climatic value (Founda et al., 2022).
Such meteorological conditions characterized by warmer than average temperatures, extremely dry air and low wind speed and



precipitation deficit served as the preconditions for the burning of the available fuel and then convert into massive wildfires. In Giannaros et al. (2022), the authors found that warmer than average temperatures and lack of precipitation catering to the two

prolonged (greater than 10 days) heat waves led to efficient drying of the fuel until the ignition time creating a highly flammable fuel. Also, the hot and dry atmospheric layer near the surface helps in maintaining intense burning as well as up–thrust of the plume.

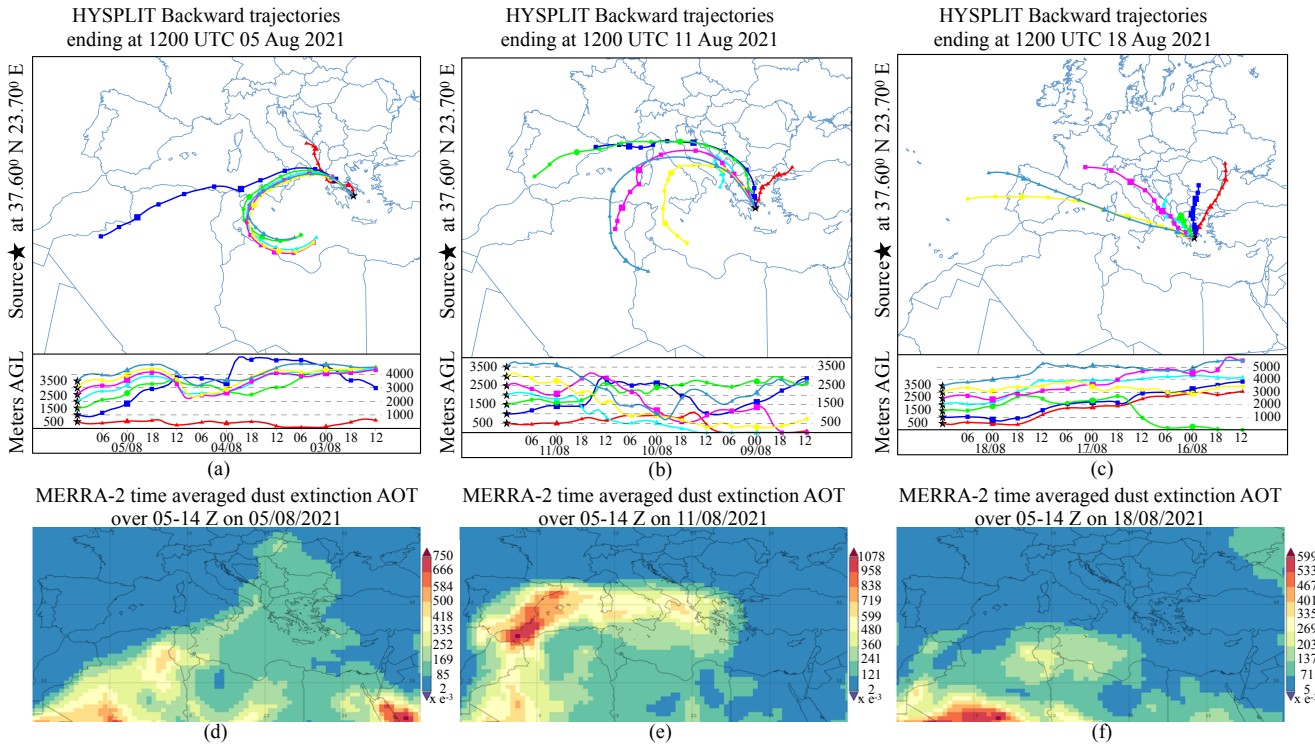

**Figure 2.** Identification of dust transfer to Athens using HYSPLIT back trajectories and MERRA–2 data.

Figure 1b presents also satellite images from MSG where the smoke plumes are evident and the sky–camera images (Fig. 1c) for Athens that confirm the presence of smoke in the region. The spread of wildfire smoke was investigated using the MSG

images (15 min frequency) while the presence of smoke over Athens was confirmed by visually inspecting sky camera images (available with a frequency of 10 s).

In addition to the prevalent smoke due to wildfires, August of 2021 also experienced episodes of dust as can be seen in the maps of the dust extinction optical thickness from MERRA–2 reanalysis images presented in Figure 2d, e and d. HYSPLIT back trajectories confirmed that the origin of air masses (and thus dust) in the particular days is the Sahara desert (Fig. 2a and

b). The Saharan dust episodes were observed on August 5 (Fig. 2a) and August 11 (Fig. 2b). According to MERRA–2 satellite images and HYSPLIT back trajectories, dusty air mass from northern Africa (Morocco, Tunisia and northern Algeria) merged over the Mediterranean Sea as a result the final air mass that arrived at Athens on 5 and 11 August 2021, included a mixture of smoke, marine and dust particles. In conclusion, the combined information of the HYSPLIT backward trajectory analysis at



the Athens station ending at 12:00 UTC, on 5 and 11 August 2021 (Figure 2) and the MERRA–2 images indicate the presence
of smoke and dust particles at altitudes below 3 km.

## 3.2 Impact on Air Quality

Both primary and secondary aerosols are produced during biomass burning whose chemical composition highly depends on
the type of combustion (flaming or smoldering) and environmental conditions (Rickly et al., 2022). The time series of the air
quality data are presented in Fig. 3, which shows the daily mean of PM2.5, PM10, NO and $NO_2$ concentrations ($\mu$g m$^{-3}$) at
various sites within GAA. Note that not all stations measure the same air pollutants, which is why the number of stations is
different in Fig. 3a–d. It was observed that PM2.5 values were generally below 20 $\mu$g m$^{-3}$ during this period except for early
and mid–August (during wildfire events) when for particular stations that were strongly affected by smoke it exceeded 70 $\mu$g
m$^{-3}$ and 60 $\mu$g m$^{-3}$ respectively (Fig. 3a). Elevated PM10 levels were also found during the same period with values reaching
up to 130 $\mu$g m$^{-3}$ (Fig. 3b). PM10 levels were maximum in the first week of August due to the presence of wildfire smoke and
desert dust over all the stations.

NOx is mainly generated during flaming stage that occurs at high temperature (Stefenelli et al., 2019). Very high NO and
$NO_2$ concentrations were also recorded in the first week of August as well as in August 18 and 19 due to the wildfire events
which is obvious as the fire events tend to increase NOx emissions (Jin et al., 2021). Daily average NO reached 30 $\mu$g m$^{-3}$
(while it is usually below 10 $\mu$g m$^{-3}$ ) while daily average $NO_2$ reached 75 $\mu$g m$^{-3}$ (while it is usually below 30 $\mu$g m$^{-3}$). But
it is interesting to note that high NO and $NO_2$ values have also been recorded in days when the aerosol mixture is constituted
mainly of dust (e.g., August 11, 25 and 26). Elevated NO/$NO_2$ levels during dust events have been also reported in other
studies (Milford et al., 2020). Increase in the total column of $NO_2$ (Fig. 3d) are generally in agreement with the increase in
surface $NO_2$ concentration. According to our analyses, increased NOx levels coincide with the presence of smoke and dust
aerosols and/or low wind speeds (see Appendix Fig. A1). The presence of dust or smoke aerosols has been reported to be
positively correlated with elevated NO and/or $NO_2$ levels in a number of studies. Low wind speed also favours increased $NO_2$
concentrations in urban environments as NOx concentrations are found to be in negative correlation with wind, precipitation
and relative humidity (Liu et al., 2020). Also, total $NO_2$ columns increased up to 6 times from the climatological mean (Fig.
A2). High values of total column $SO_2$ were also observed during the first week of August (highest on August 7) and then
later on August 19 with values reaching as high as 8 DU and 6 DU, respectively (Appendix Fig. A2) while the climatological
average is ∼1 DU. During wildfire events unusually large amounts of $SO_2$ have been observed in previous studies including
(Rickly et al., 2022; Weber et al., 2021; Ren et al., 2021).



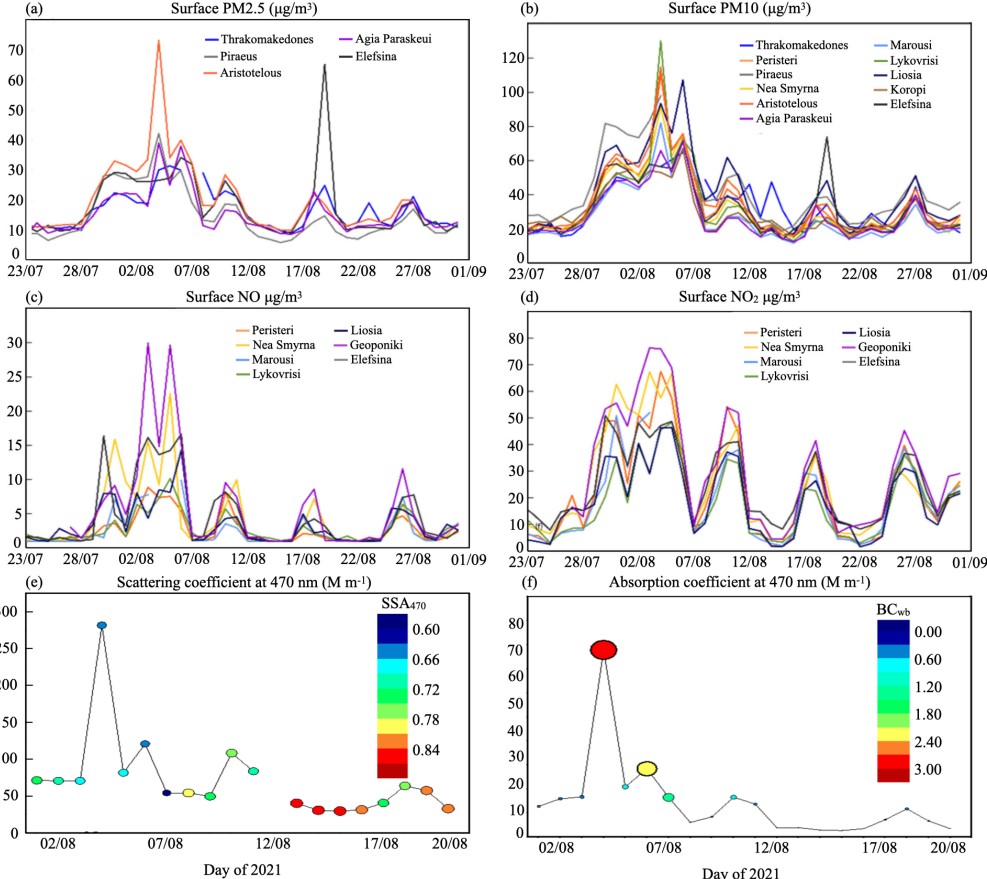

**Figure 3.** Variation of (a) PM2.5 ($\mu$g m$^{-3}$), (b) PM10 ($\mu$g m$^{-3}$), (c) NO ($\mu$g m$^{-3}$) and (d) NO$_2$ ($\mu$g m$^{-3}$) in Greater Athens Area (GAA) during August 2021. Temporal variation of the daily-mean values for the scattering (e) and absorption (e) coefficients in Athens during 1-21 August 2021. The data points in (e) and (f) are color-coded as a function of SSA$_{470}$ and BC$_{wb}$ concentration, respectively.

In Wu et al. (2021) on wildfire, PM2.5 and organic carbon showed a sharp increase (PM2.5 were 5 $\mu$g m$^{-3}$ before the wildfire and 30 $\mu$g m$^{-3}$ after) signifying that the air quality is affected by the transport of wildfire smoke. The daily evolution of the aerosol scattering coefficient (b$_{sca}$,470) clearly detects the effect of Attica forest fires on the light scattering (Fig. 3e),

with daily–mean b$_{sca}$ value of 282 M m$^{-1}$ on August 4, and enhanced (> 100 M m$^{-1}$) b$_{sca}$ values on other days significantly affected by transported smoke plumes over Athens like 6 and 10 August. The mean SSA660 during the measuring period was 0.77 (0.02 higher than SSA470), while under intense smoke conditions (August 4), this difference increased to 0.05 (SSA660 = 0.70), suggesting enhanced presence of brown carbon (BrC) aerosols. The peak values of b$_{abs}$ on August 4 and August 6, associated with higher BCwb concentrations are characteristic of the strong smoke effect on light absorption, while this effect

was much more intense at 370 nm (b$_{abs}$,370 = 156.7 M m$^{-1}$ on August 4). The BCwb concentrations in August 2021 (0.43





± 1.21 $\mu$g m$^{-3}$) was much higher — and variable as well — than the 4–year August mean value of 0.22 ± 0.20 $\mu$g m$^{-3}$ (Liakakou et al., 2020).

## 3.3  Aerosol optical and microphysical properties

Figure 4a shows the variation of the AOD at five wavelengths namely 340 nm, 440 nm, 675 nm, 870 nm and 1020 nm (from
now on referred as C5) and the Ångström exponent (AE) at 440–870 nm while Fig. 4b presents the variation in the fine mode, coarse mode and total AOD at 500 nm and the fine mode fraction during August 2021. During August 2021, the mean AOD was found to be 0.462, 0.352, 0.206, 0.153 and 0.131 at C5. The fine mode AOD was found to be as high as 1.95 on August 18 followed by 1.49 on August 7, 1.21 on August 5, 0.99 on August 4, 0.96 on August 8 and 0.86 on August 9 and the corresponding fine mode fraction on these days being 0.98, 0.97, 0.85, 0.99, and 0.97, respectively indicating high dominance
of fine particles. On the contrary, the fine mode fraction on August 11 was observed to be 0.31 with fine mode and coarse mode AODs being 0.34 and 0.74, respectively indicating the dominance of coarser particles probably due to dust activity. The coarse mode AOD on August 7, August 8, August 9 and August 18 was 0.04, 0.01, 0.03 and 0.05, respectively indicating that in these days smoke was mainly present. While on August 4 and August 5, the coarse mode AOD was up to 0.22 and 0.25, respectively indicating that these days have both the presence of dust and smoke. The detailed values of aerosol properties on these days
are presented in Appendix Table A1.

Figure 4c presents the volume particle size distribution and variation of single scattering albedo during August 2021 in Athens which are produced using the daily averages of 22 logarithmically equidistant discrete points in the size range varying from 0.05 $\mu$m to 15 $\mu$m. The variation of single scattering albedo is presented in Figure 4d at 440 nm, 675 nm, 870 nm and 1020 nm. There were six interesting cases observed in Athens (with AOD values equal to 1 or more) on August 5, 7, 11, 17,
18 and 19. The smoke plume on August 7, after transported, was detected from the Polly$^{XT}$ lidar above PANGEA (discussed in Section 3.5). Smoke over PANGEA was observed in altitudes between 0.5 - 3.5 km above the surface. Figure 5 shows the time-height distribution of aerosol layers in the atmosphere using the attenuated backscatter collected by the ceilometer at Athens during the wildfire event of August 2021. The most markable aerosol layers were observed from August 4 to 7, August 11 and from August 17 to 18 as can be seen from Figure 5. Henceforth, we consider three cases as case 1 with only smoke
event, case 2 with only dust activity and case 3 with both dust and smoke activities. It can be observed that August 15, 16, 28 are characterized by very low aerosol load and hence can be used as reference cases for Athens. The variation in the aerosol properties in these three cases are presented in Table 2.





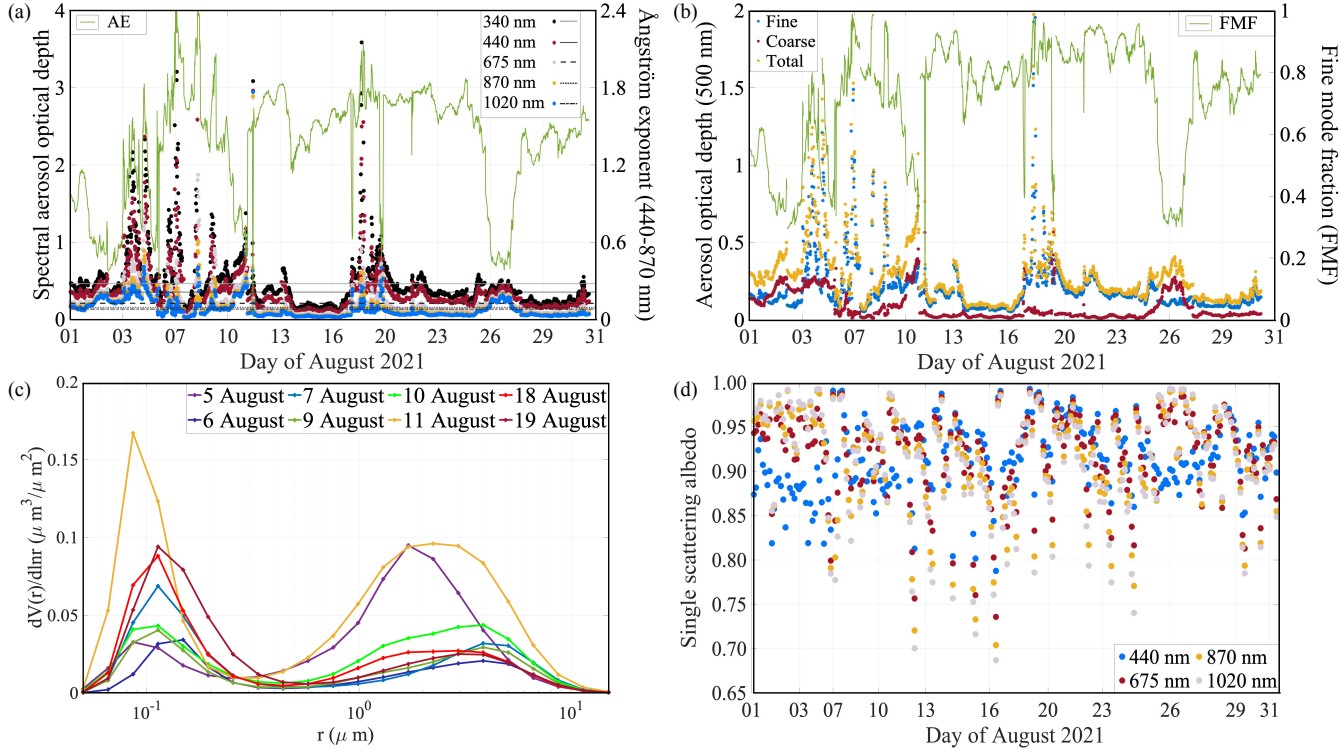

**Figure 4.** Variation of (a) Aerosol spectral optical properties and (b) fine and coarse mode AOD at 500 nm (c) daily mean volume particle size distribution and (d) single scattering albedo during the wildfire event in Athens

**Table 2.** Average aerosol properties (maximum values in bracket) for smoke and/or dust events of August 2021

|  | Event | AOD500 | Fine AOD500 | Coarse AOD500 | FMF500 | AE 440–870 | SSA (440–1020) |
|---|---|---|---|---|---|---|---|
| Case 1 | Smoke | 0.57 (1.53) | 0.50 (1.95) | 0.06 (0.17) | 0.85 (0.99) | 1.84 (2.41) | 0.93–0.86 (0.99–0.95) |
| Case 2 | Dust | 0.57 (1.07) | 0.26 (0.34) | 0.31 (0.74) | 0.46 (0.54) | 0.72 (0.89) | 0.89–0.97 (0.90–0.99) |
| Case 3 | Dust & smoke | 0.63 (1.43) | 0.39 (1.21) | 0.23 (0.28) | 0.57 (0.85) | 1.46 (1.56) | 0.87–0.97 (0.90–0.99) |




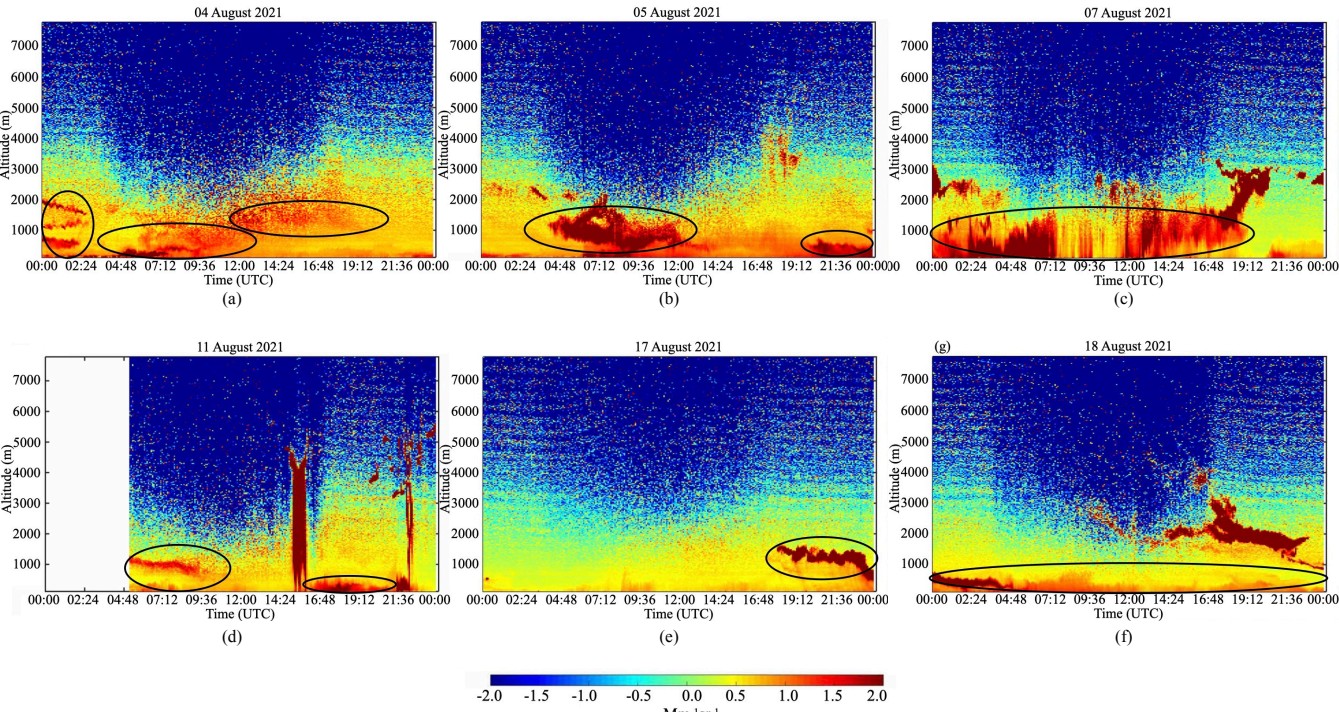

**Figure 5.** Time-height distribution (a, b, c, d, e, f) of ceilometer attenuated backscatter coefficient at Athens between August 5 and August 20, 2021. Black circles represent the smoke layers and not encircled red features are clouds and/or rain.

**Case 1: Smoke**

August 7, 17, 18 and 19 depicts the presence of only smoke. August 7 was characterized by the highest AOD values of the month varying from 3.20 to 0.34 at C5 with the mean and maximum AE being 1.95 and 2.41, respectively. The mean values of fine mode AOD, coarse mode AOD, total AOD at 500 nm and fine mode fraction on 7 August were found to be 0.51, 0.06, 0.58 and 0.87, respectively with their maximum values being 1.49, 0.17, 1.53 and 0.99, respectively. The high values of AE denotes the dominance of small smoke particles in the aerosol mixture. August 17 was signified by maximum AOD values between 0.78 and 0.39 at C5 with the mean and maximum AE being 1.53 and 1.79, respectively. This day had the mean fine mode, coarse mode and total AODs at 500 nm as 0.10, 0.04 and 0.15, respectively with the corresponding maximum values being 0.22, 0.33 and 0.55, respectively. August 18 was characterized by maximum values of AODs between 3.59 and 0.42 at C5 and the AE showed a mean and maximum value of 1.73 and 2.13, respectively. This day had an average values of fine mode, coarse mode and total AODs at 500 nm as 0.49, 0.06 and 0.56, respectively while the maximum values went up to 1.95, 0.08 and 2.00, respectively. Finally, on August 19, the AOD values reached to a maximum between 1.31 and 0.66 at C5 and the AE values reached as high as 1.98 with an average value of 1.54.



A strong absorption characteristic and strong spectral dependence is observed on August 7 when the SSA is seen to mono-tonically decrease with wavelength from 0.93 at 440 nm to 0.86 at 1020 nm. Similarly, the SSA is seen to decrease from 0.92 to 0.87 from 440 nm to 1020 nm on August 17 indicating the presence of fresh smoke (Reid and Hobbs, 1998; Dubovik et al., 2000), while on August 19, the SSA decreases from 0.95 at 440 nm to 0.92 at 1020 nm, however the decrease is not as

prominent as on August 17 signifying the presence of residue smoke as they tend to be slightly less absorbing (Gómez-Amo et al., 2017). It is also observed that the SSA reaches very low values (even below 0.7) at 1020 nm during smoke events indi-cating the presence of strong absorbing aerosols. In Wu et al. (2021), NOAA hazard mapping system and HYSPLIT backward trajectories were used to study the source and transport of the wildfire and lidar–ratio was used for distinguishing smoke par-ticles from the urban aerosols (larger lidar–ratio signifying the presence of smoke). The extinction Ångström Exponent from

AERONET in near–infrared (NIR) and ultraviolet (UV) wavelengths were used to analyze the smoke loadings and was found to be correlated to the smoke AOD. Also, it was observed that the contribution of smoke to the AOD was about 60–70 % and the presence of black carbon, ozone and carbon monoxide was observed in the elevated smoke layers as obtained from aircraft in situ observations.

A high intensified aerosol layer is observed below 2 km altitude on August 07 as it appears from Figure 5c which persists

for the entire day. Moreover, August 17 displays a fairly stable atmospheric composition as can be seen from Figure 5e, but a dense afloat aerosol layer can be seen after 19:00 UTC that descends down from 2 km altitude at 19:00 UTC to below 1 km at mid-night. This aerosol layer remains there till 4:00 UTC on August 18 as can be seen from Figure 5f and it mixes up in the boundary layer afterwards. However, another dense floating aerosol layer can be seen after 14:00 UTC above 2 km altitude which stays in the boundary layer till night.

**Case 2: Dust**

August 11 had the presence of dust. On August 11, the AOD at C5 reached maximum values of 1.37, 1.17, 0.94, 0.87 and 0.86, respectively and the AE displayed an average and maximum value of 0.72 and 0.89, respectively. The average fine mode AOD, coarse mode AOD, total AOD at 500 nm and fine mode fraction were estimated to be 0.26, 0.31, 0.57 and 0.46, respectively with the highest values reaching 0.34, 0.74, 1.07 and 0.54, respectively. Low AE on August 11 indicates the presence of larger

particles (Pace et al., 2006) which, as can be perceived by Fig. 2 are dust particles that have been transported to Athens. From Fig. 4d (also Appendix Table A1), the SSA on August 11 is seen to increase with wavelength from 0.89 at 440 nm to values above 0.95 at wavelengths between 675 nm and 1020 nm which signifies large forward scattering due to the presence of dust particles (Gómez-Amo et al., 2017). This is a typical spectral behavior of dust aerosols having more absorption in UV than in near infrared (Dubovik et al., 2002; Derimian et al., 2008). From Fig. 5d, it is seen that on August 11 there is a floating dust

aerosol layer around 1 km altitude till 10:00 UTC.

**Case 3: Smoke and dust**

August 4 and 5 were characterized by the presence of both dust and smoke. On August 5, the maximum AOD values at C5 were found to be 2.33, 2.36, 1.41, 0.89 and 0.68, respectively. The average and maximum AE at 440–870 nm were found to



be 1.04 and 1.82, respectively. The large difference in the average and maximum value of Ångström exponent indicates that
there was a drastic variation in AE during this day. It was found that the AE varied from 1.82 in the morning to about 0.53 in
the evening. High AOD and high AE in morning and high AOD and low AE in evening indicate that the aerosol mixture in the
morning was dominated by small smoke particles while in the evening it was dominated by large dust particles (Gómez-Amo
et al., 2017). The average values of fine mode AOD, coarse mode AOD, total AOD at 500 nm and fine mode fraction were 0.36,
0.23, 0.59 and 0.53, respectively while their respective maximum values were 1.21, 0.27, 1.43 and 0.85, respectively. Figure 2a
indicates the transport of Saharan dust to Athens thus signifying the presence of both smoke and dust on August 5. August 4
had mixed aerosol in the boundary layer below 2 km (Fig. 5a). A dense floating layer of aerosol is observed from Figure 5b on
August 5 at about 1 km altitude and mostly below 2 km around 7:00 UTC. During the nocturnal hours, the highlighted aerosol
layers are observed below 1 km altitude.

## 3.4 Aerosol properties from CAMS and SKYCAM

Figure 6a shows the total, organic matter and dust AODs from CAMS. It is observed that the organic matter AOD is highest
on August 7 while the values also peak between August 17 to August 19. While the dust AOD peaks between August 1 and
August 6, August 11 and August 27 and is nearly negligible between August 13 to August 24. Figures 6b and 6c compare
the daily average AOD from CIMEL with that from CAMS and SKYCAM retrievals. For the comparison between the daily
average AOD from CAMS and CIMEL shown in Fig. 6b, the AOD from CAMS at 550 nm has been extrapolated to 500 nm
using the daily average AE from CIMEL. The daily average AOD at 500 nm from the SKYCAM has been also compared with
the corresponding AOD from CIMEL. It must be noted that the SKYCAM retrievals used for the calculation of daily averages
were simultaneous with the CIMEL AOD retrievals. For most days, SKYCAM gives higher values than the CIMEL with the
difference between the AOD from CIMEL and the SKYCAM being less than 0.1 (Fig. 6c). However, during intense smoke
events differences are much larger, up to 0.2. Differences between the AOD from CAMS and the AOD from CIMEL are also
below 0.1 with the exception of days with dust or smoke events. Dust AOD is generally overestimated by CAMS (e.g., by 0.2
on August 11). During smoke events differences between CAMS and CIMEL are again larger.





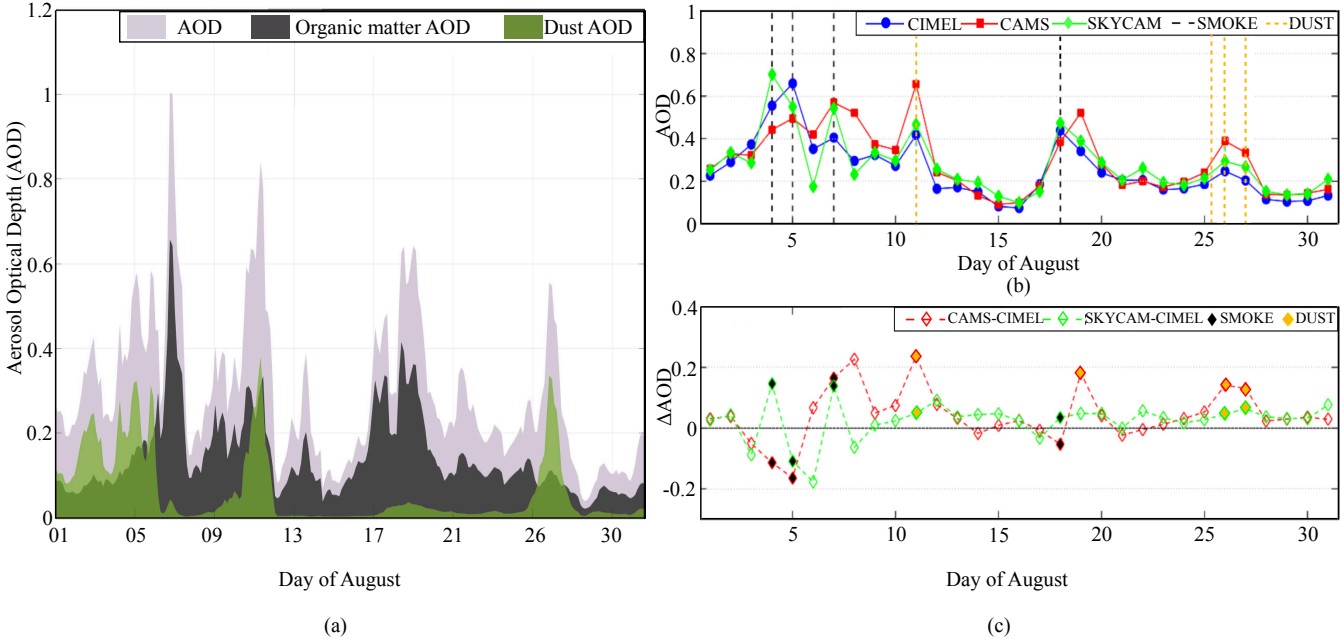

**Figure 6.** (a) Total, organic matter and dust AOD from CAMS at 550 nm. (b) Comparison of AOD from CIMEL with AODs from CAMS and SKYCAM at 500 nm (see text). (c) AOD differences between CAMS and CIMEL, and between SKYCAM and CIMEL at 500 nm.

## 3.5 Transformation during transport over PANGEA

The smoke plume on August 7, after transported, was detected from the Polly$^{XT}$–NOA lidar above PANGEA. Smoke over PANGEA was observed in altitudes between 0.5–3.0 km above the surface. Fig. 7 shows the Polly$^{XT}$–NOA lidar attenuated

backscatter coefficient at 1064 nm and the 3–day air masses back trajectories above the station (ending at 12:00 UTC on August 7, 2021) from FLEXPART–WRF and HYSPLIT model simulations. On August 7, there was the transfer of smoke over PANGEA from Athens as can be seen from Fig. 7b and c. Wildfire aerosol sources and transports, lidar measurements and analyses with different models confirm that the smoke from Athens has been transferred to PANGEA as can be seen from the layer at 1–2 km in Fig. 7a. and the time needed for the transfer was between 4 to 9 h. In Castagna et al. (2021), the authors

used satellite and ground–based fire data to run the WRF-HYSPLIT model and found that out of the total wildfire cases, 52.5 % were located outside the Calabria Region, impacted by long–range transport.



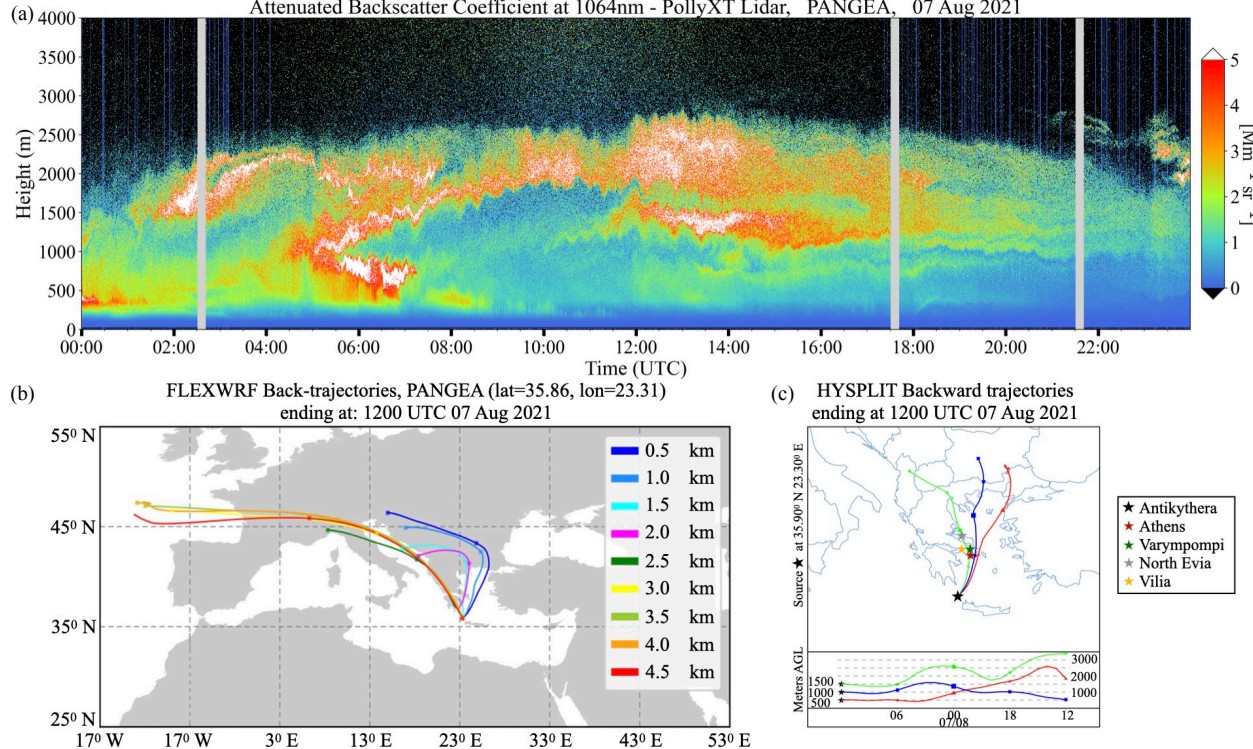

**Figure 7.** (a) Lidar attenuated backscatter coefficient at 1064 nm on August 7 (b), FLEXPART–WRF 3–day back trajectories of air masses at 12:00 UTC (each color corresponds to the trajectories ending at 0.5, 1, 1.5, 2, 2.5, 3, 3.5, 4 and 4.5 km above ground level) and (c) HYSPLIT 1–day backward trajectories ending at 12:00 UTC on August 7, 2021 over PANGEA.

In order to further analyse the transfer of smoke to PANGEA, a comparison between the variation in total AOD, fine AOD and coarse mode AOD at 500 nm, and Ångström exponent (440–870 nm) at PANGEA and Athens was carried out for August 7 as is presented in Fig. 8a and Fig. 8b, respectively. Moreover, Fig. 9 shows the single scattering albedo for PANGEA and

Athens at 440 nm, 675 nm, 870 nm and 1020 nm. Figure 9 has been created from averages for August 7 when smoke was present over Athens and over PANGEA. It is observed that SSA and AE changed during the transfer from Athens to PANGEA. An impressive change in the spectral shape of the SSA can be observed from Fig. 9 given that transfer of smoke from Athens to PANGEA took place in less than 9 hours. The median SSA value at PANGEA is observed to decrease monotonically from 0.96 at 440 nm to 0.93 at 1020 nm with the values at 675 nm and 870 nm being 0.95 and 0.94, respectively. At Athens, the median

SSA value was found to have a more drastic decrease from 0.90 at 440 nm to 0.80 at 1020 nm with the values at 675 nm and 870 nm being 0.86 and 0.82. The decreasing SSA value with wavelength indicates the presence of smoke (Gómez-Amo et al., 2017) which is evident for both the stations. But the spectral curve of the two station signifies that the smoke aged and the plumes diluted during the transport from Athens to PANGEA. A probable explanation for this phenomenon could be aerosol removal due to dispersion, coagulation and sedimentation and decrease in light scattering efficiency with distance and time

(Radke et al., 1995). The aging of the smoke plume leads to coagulation of the particles in the accumulation mode and shift to





coarse mode with time as presented in Radke et al. (1995) where the authors found that spherical particles of 2 PM diameter falls tens of meters a day. But this change took place in times of the order of a few days (in 50 h as the smoke aged). However, in the study presented here, this change happened in only a few hours. Hence, sedimentation can be another factor in removal of the aerosol particles. Furthermore, there can also be an increase in the effective density of smoke particles due to decrease in

drag with time as a result of coagulating particles developing into more spherical aggregates (Friedlander and Marlow, 1977). Another reason for a more flatter SSA curve for PANGEA could be the presence of marine aerosols as marine SSA spectral dependence is more flat. The AE also drops slightly in PANGEA than in Athens (Fig. 8) indicating the contribution of larger particles to the column like marine aerosols. In Gómez-Amo et al. (2017), the authors found that the wildfire related smoke event and a dust episode were simultaneously detected and the dust-smoke mixing was found to enhance the aerosol load and

modify the aerosol properties. The AOD was found to increase up to 1 due to dust and to an extreme of 8 as a consequence of smoke. The bimodal size distribution of the mixture was found to be dominated by smoke and dust in fine and coarse modes, respectively.

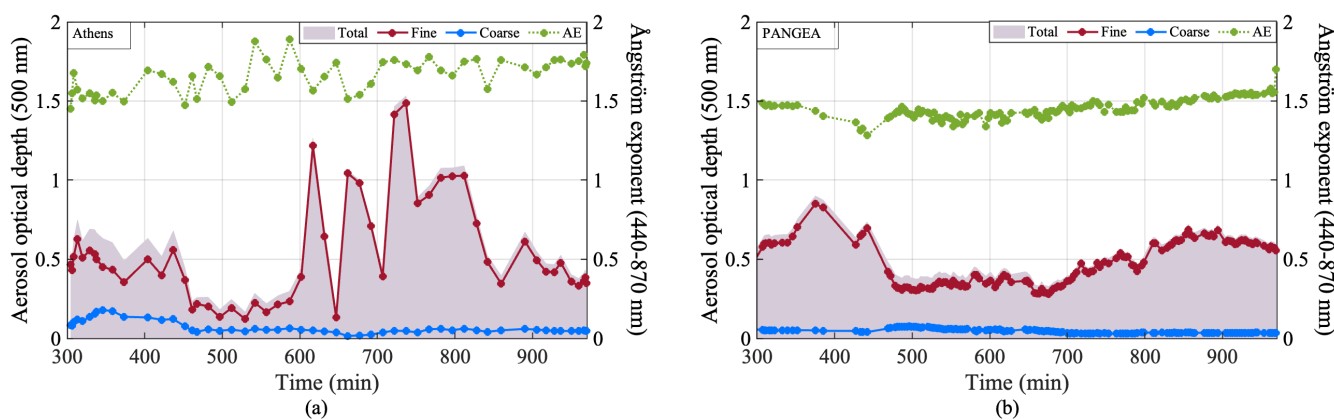

**Figure 8.** Variation of total AOD, fine AOD and coarse mode AOD at 500 nm, and Ångström exponent (440-870 nm) at (a) Athens and (b) PANGEA during the wildfire event of August 7, 2021





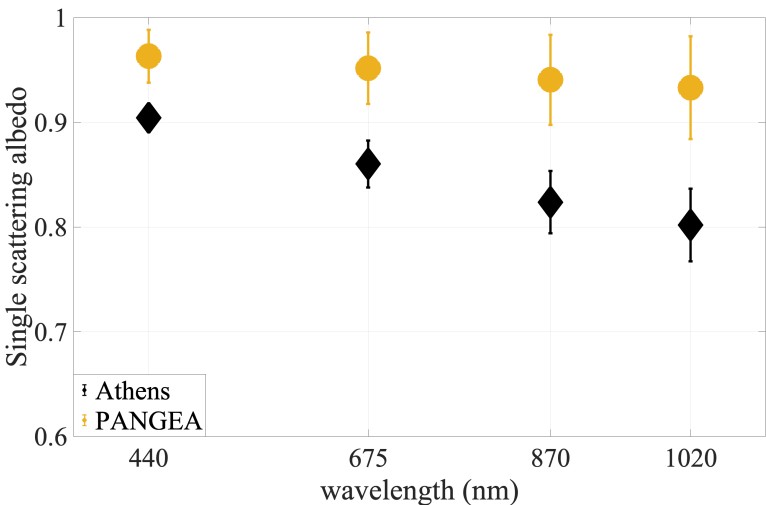

**Figure 9.** Single scattering albedo for Athens and PANGEA during the wildfire event of August 7, 2021

## 3.6 Effect on solar radiation

### 3.6.1 Spectral and total solar radiation

Finally, we calculated the attenuation by dust and smoke in different spectral regions during specific high–AOD days of August 2021. For this purpose, we compared measured irradiances at different spectral bands with the corresponding modelled irradiances for aerosol–free skies (case (c) in Section 2.3.1). In order to ensure that the modelled and the measured irradiances are comparable, we also modelled the irradiances using CIMEL measurements (case (a) in Section 2.3.1) and then compared measured and modelled irradiances for days with very low aerosol load. When AOD is low, uncertainties in the aerosol optical

properties used for the simulations have a negligible impact on the simulated irradiances. The lowest AOD–days were the 15th and the 16th of August as inferred from Section 3.3. The results for both days were nearly identical and yielded an agreement better than 2% between the measured and modelled irradiances for SZAs below 80°. For the 16th of August the ratio between the measured and modelled (considering realistic aerosol conditions) is presented with dotted lines in Figure 10o. The results of the comparison between measured and modelled (considering AOD=0) irradiances for 4 different days are also presented

in Figure 10 for UV–B (10a–d), VIS (10e–h), and NIR (10i–l). These days are chosen as representative for different events as presented in Section 3.3. This includes August 4 with very high AOD due to the presence of both, dust and smoke, August 11 with very high AOD due to the presence of dust, August 16 representing very low AOD (daily average below 0.05 at 500 nm) and August 18 with very high AOD due to the presence of smoke.



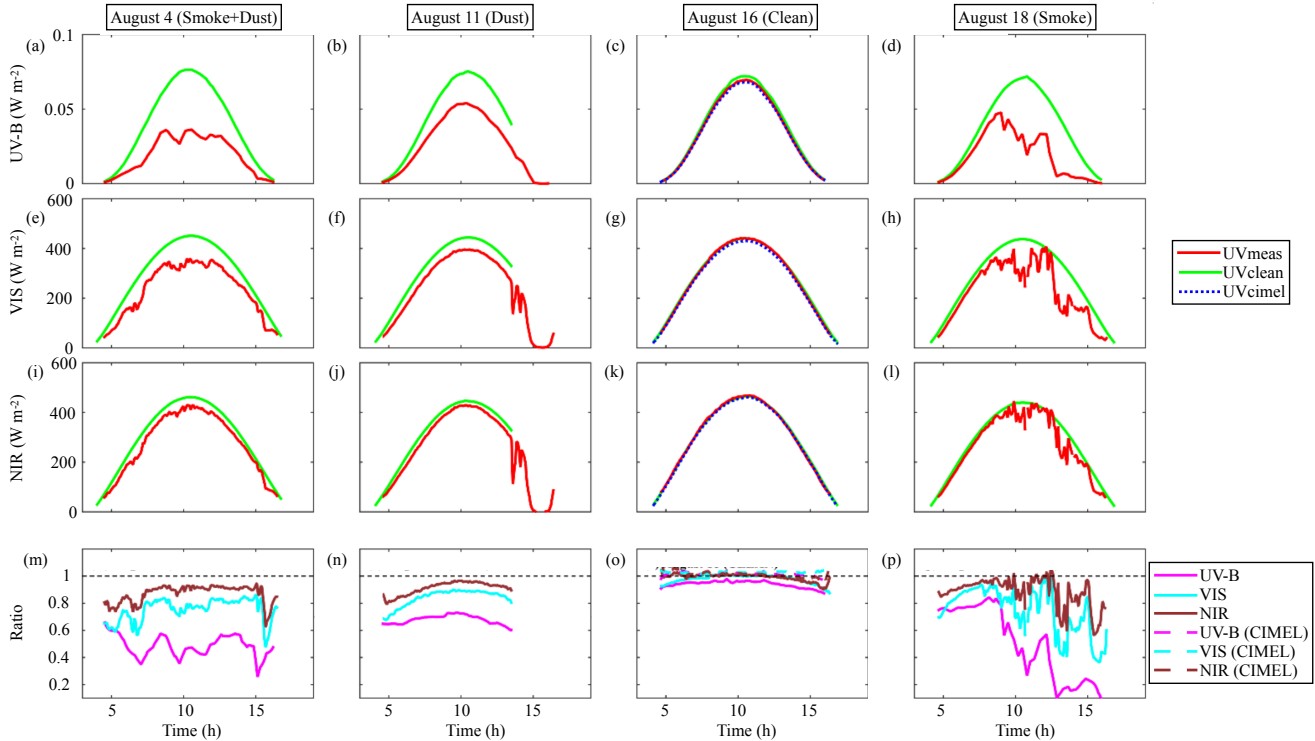

**Figure 10.** Effect of smoke and dust aerosols on UV–B (a, b, c, d), VIS (e, f, g, h) and NIR (i, j, k, l) irradiance on August 4, August 11, August 16 and August 18, respectively and the ratio (m, n, o, p).

From Figures 10a–d, it is observed that the attenuation of UV–B irradiance was least on August 11 and it was highest on
August 18 followed by August 4. It is to be noted that August 4 and August 18 are the days corresponding to smoke aerosols with very high fine mode AOD values (>1) as presented in Section 3.3 while August 11 has low fine mode AOD but high coarse mode AOD. Also, the very high AE of smoke, combined with the low SSA induce a steep gradient in the spectral dependence of the attenuation. Thus, on August 4 and 18, the UV–B irradiance was attenuated by 60 %. Moreover, in the evening of August 18, the smoke aerosols attenuated about 90 % of UV–B irradiance or more. Moreover, the attenuation in
NIR was comparatively less as can be seen from Figures 10i–l which was mostly of the order of 20% or less. However, the attenuation of NIR irradiance was greater in the evening of August 18, as was the case with UV–B irradiance, reaching about 40 %.

Figure 11 shows the relative contribution of the different spectral regions (UV–B, UV–A, VIS, NIR) to the daily integrals of the GHI irradiance. The contribution is calculated as the ratio between irradiance in a spectral region (NIR, VIS and UV)
to the GHI. Due to relatively large gaps in the Brewer measurements in 7 and 12 of August the UV–B integrals have not been calculated for these days. The theoretical integrals that have been calculated based on modelled irradiances are presented with dashed lines.



Figure 11a shows the contribution (proportion) of visible and NIR to total irradiance. It can be observed that the contribution of NIR to total irradiance is higher on smoke days than in dust days while the opposite can be observed for the VIS range.

Figures 11b and c show the contribution of UV–A and UV–B to total irradiance. As in the VIS range the contribution from UV–A and UV–B is lower for smoke cases as expected, due to the higher spectral dependence (AE) of AOD to smoke aerosols being higher in the lower spectral ranges. The daily average AOD at 500 nm and at 340 nm is shown in Appendix Table A1. It is interesting that although the daily average AOD at 500 nm is the same, equal to 0.58 on August 7 and August 11, the change in the composition of GHI is significantly more pronounced on August 7 because of the larger AE in this date. The explanation

is the much larger AE of smoke (values of ∼2 were measured on August 7) relative to the AE of dust (values of ∼0.6 were measured on August 11).

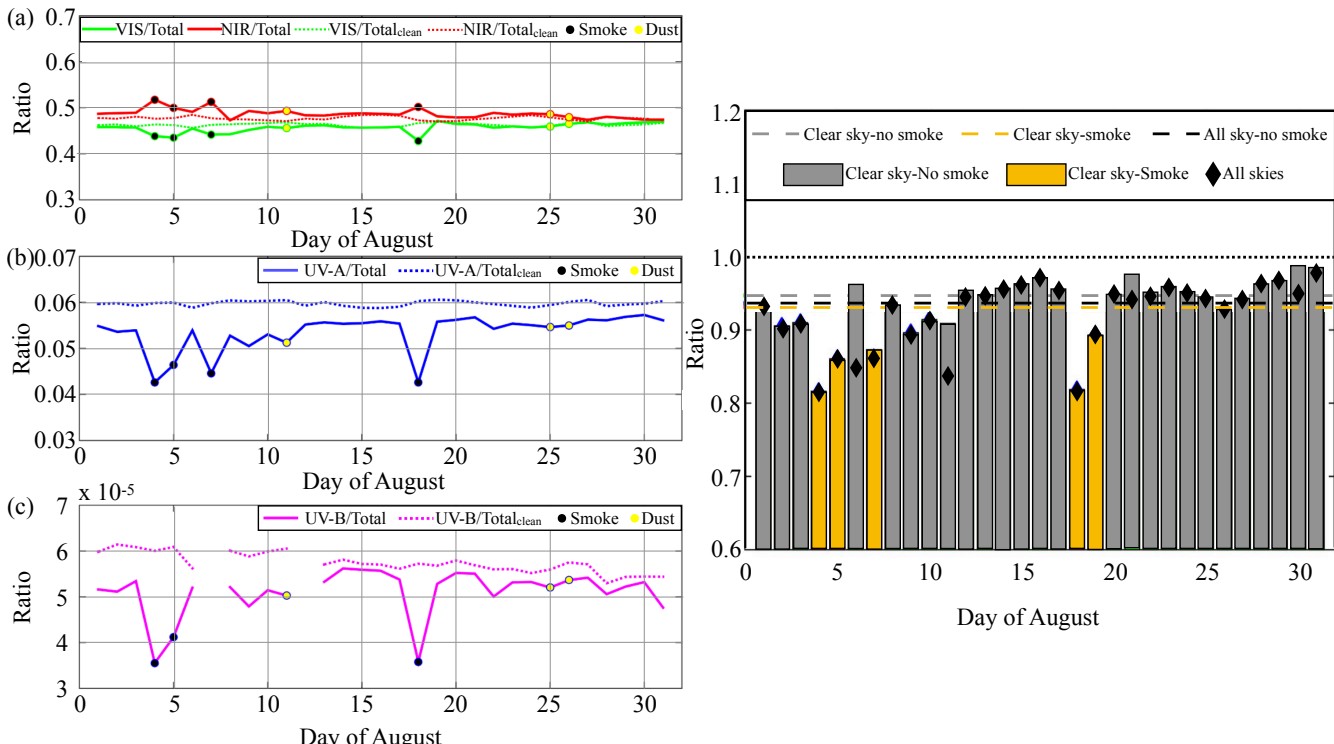

**Figure 11.** Contribution of different spectral regions to total solar irradiance in August 2021. Panel (a): Ratios of NIR and VIS with GHI, panel (b) ratio of UV–A with GHI, (c) ratio of UV–B with GHI (d) Effect of smoke on the levels of GHI. The dashed lines (gray, gold and black) represent the ratio based on average GHI for August while the ratio based on daily GHI are represented by gray and gold bars and black rhombus. Continuous lines represent ratios calculated using measured values, while dotted lines represent ratios calculated using modelled values for aerosol free conditions. Black dots represent smoke events while yellow dots represent dust events.

The effect of smoke on the levels of daily and monthly GHI in August 2021 is presented in Figure 11d. The ratios presented in Figure 11d have been calculated as the ratio of the daily integrals from the pyranometer measurements divided by the daily



integrals calculated from modelled irradiances for AOD=0 and are represented by the black rhombus. In order to exclude the effect of clouds there was a visual inspection of the measurements with respect to cloud camera images and the hours during which the sun disk was partially or fully covered by clouds were marked. Then, for these hours the modelled irradiances were assumed to be equal to the measured irradiances (assuming that the aerosol effects are negligible under cloudy conditions). Measurement–based integrals were then divided with the latter modified modelled integrals (gray and gold bars). Intense smoke events were marked with gold color. Black dashed line represents the ratio between the average of measurement–based daily integrals and the model–based daily integrals, excluding the days corresponding to intense smoke events. The gray dotted line represents the ratio between the average of measurement–based daily integrals and the modified model–based daily integrals for clear sky days, excluding again the days corresponding to intense smoke events. The ratio represented by the gold dashed line has been calculated the same way (as ratio represented by the gray dashed line) including the days with intense smoke events.

During intense smoke events the daily GHI was attenuated by 10–20 % leading to a decrease of ~1.5 % in the monthly GHI. If days with smoke are not taken into account, the overall GHI decrease due to aerosols is ~5.5 % (gray dashed line). By taking the effect of clouds into account for the same days (black dashed line) the decrease is 6.5 %. If only cloudless conditions are considered including intense smoke events the overall monthly GHI attenuation is 7 % (gold dashed line). In August energy demand is high due to high temperatures, especially under extreme heatwave events (such as the one in August 2021). In a future where significant fraction of consumed energy will emerge from photovoltaics, decreases in GHI of 10–20 % could have a significant impact on many human activities that are strongly related with solar energy production.

### 3.6.2 Biologically effective quantities

The effect of the intense smoke events on the levels of different biologically effective doses was also investigated. For this part of the study the measured doses were compared with modeled doses that were calculated for climatological aerosol optical properties (case (b) of Section 2.3.1). In Fig. 12, the vitamin D and PAR daily doses and the maximum UV index are presented as they were calculated from Brewer#001 (maximum UV index and daily vitamin D) and PSR measurements and from libRadtran simulations. The corresponding ratios between measured and modelled doses are also presented. The maximum UV index was calculated as the average UV index within ± 30 minutes around the local noon.

Despite the distance of 6 km between the PSR and the Brewer#001 the calculated doses from the two instruments agree quite well, within less than 5 % during the dust and smoke events, confirming that the effects of dust and smoke aerosols were quite homogeneous over the city center during the events. The presence of smoke at the noon of the 4th and 18th of August resulted in UV indexes that are correspondingly 4.5 and 2 units below the climatological levels. Decreases of 30–50 % in the daily vitamin D doses were also estimated for extreme-smoke events. Attenuation of the daily vitamin D dose on the 7th of August is 40–50 % and thus nearly double than the attenuation by dust on August 11 (~20 %) although the daily average AOD at 500 nm was larger on August 11 (see Section 3.6.1).



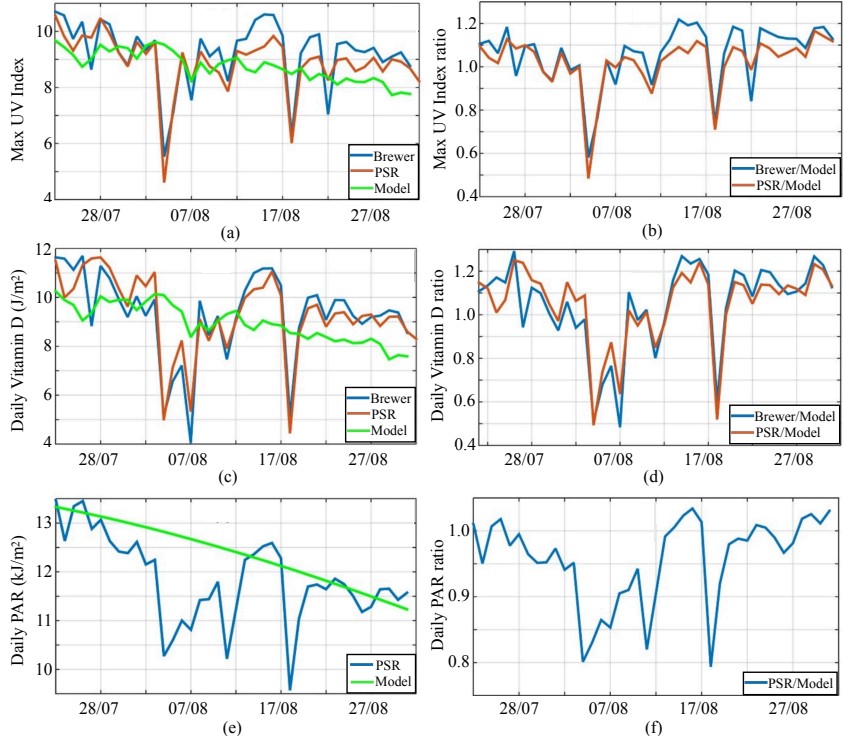

**Figure 12.** Variability in biologically effective doses from Brewer, PSR and libRadtran simulations for climatological aerosol optical properties (panels a,c,e) and the corresponding ratios between Brewer and modeled and PSR and modeled doses (b,d,f).

In a study of the wildfire event by (Gómez-Amo et al., 2019) the impact of the event on photovoltaic plant performances was studied by analyzing the radiative effects of smoke and dust. It was found that there was a loss of energy due to smoke with an average of 34% daily while due to dust it was around 6%, signifying the much higher efficiency of smoke in diminishing the energy generation as compared to dust. In Filonchyk et al. (2022), it was presented that the AOD and Ultraviolet Aerosol Index

(UAI) was found to exceed 1 and 2, respectively in general and in some parts even reaching up to 3.7 and 6.6, respectively followed by a wildfire event. Our findings on the environmental and atmospheric impacts associated with large forest fires are in line with the results of these studies.

## 4 Summary and conclusions

Significant impact of severe forest fires on air quality and solar irradiance was observed in Greece in August of 2021. The

AOD concentrations increased up to 12 times and total $NO_2$ up to 6 times higher from the climatological mean. Total $SO_2$ reached as high as 8 DU while the climatological average is about 1 DU. Significant elevated levels were also recorded in surface PM2.5, PM10, and $NO/NO_2$ concentrations. In situ aerosol measurements showed that the transported smoke plumes over the Athens urban environment also exhibited a large effect on in situ aerosol measurements by increasing signficantly the





scattering and absorption coefficients near the ground, as well as the AE values, with a concurrent decrease of the SSA470 at
about 0.65–0.70. Furthermore, the forest fires highly increased the BC concentrations in Athens, and especially the component
related to biomass burning (BCwb), which in August 2021 was double than the long-term August value.

Wildfire smoke was also observed to be accompanied by the Saharan dust on few days in August. Based on the AOD, AE,
volume size distribution, spectral variation of SSA and on the synergistic use of the ceilometer vertical distribution, it can be
inferred that August 4 and 5 were characterized with the presence of both dust and smoke, while August 7, 17, 18 and 19
depicts the presence of only smoke and August 11 had the presence of dust. Only dust days were found to have high AOD, low
AE and positive dependence between SSA and wavelength indicating large forward scattering due to coarse particles. While
the days with the presence of only smoke had high AOD, high AE and a negative dependence between SSA and wavelength.
Also, days with fresh smoke had stronger spectral variation in SSA as compared to aged smoke. Separate analyses of total
AOD, organic matter AOD and dust AOD from CAMS showed similarly the presence of high organic matter on only smoke
days (August 7, August 11 and between August 17 to August 19) and peak dust AOD on August 11.

On August 7 the smoke plume travelled from Athens to PANGEA, which is about 240 km from Athens, in about 4–9 h.
The transport of the plumes was detected using HYSPLIT and WRF–FLEXPART backward trajectories ending at PANGEA,
originating from Athens. Further, a significant change in the smoke properties was observed during this transport during which
SSA and AE changed. Most importantly, there was an impressive change in the spectral shape of the SSA. At Athens the SSA
monotonically decreased from 0.9 to 0.8 with an increase in wavelength from 440 nm to 1020 nm. While at PANGEA, this
decrease was comparatively less (from 0.96 to 0.93). Hence, the spectral curve SSA of the two stations signified that the smoke
aged, and the plumes diluted during the transport from Athens to PANGEA.

Further the attenuation of solar irradiance in different spectral regions due to the presence of dust and smoke was analyzed.
It was found that the attenuation of UV–B irradiance was least in the presence of dust and highest due to smoke (up to 60 %
or more) and intermediate when there was a mixture of smoke and dust. However, the attenuation in NIR was comparatively
less and mostly of the order of 20 % or less but the attenuation even reached up to 40 % in the presence of smoke. In VIS
region, the attenuation was greater than NIR region but less than that in UV–B region. The relative contribution of the different
spectral regions as compared to the daily integrals of the GHI irradiance was also analyzed and it was found that the higher
spectral dependence of AOD for smoke particles leads to lower relative contributions to lower wavelengths (UV, VIS) and
higher relative contributions to the NIR, compared to the ones for the dust cases.

The effect of smoke on the levels of daily and monthly GHI was also considered and it was observed that during intense
smoke events the daily GHI got attenuated by 10–20 %. However, during the absence of smoke, the overall GHI decrease
due to aerosols was ∼5.5 %. Also, when clouds were taken into account, the decrease was found to be 6.5 %. Furthermore,
when only cloudless conditions were considered along with intense smoke cases, then the overall monthly GHI attenuation
was found to be 7 %. In August energy demand is high due to high temperatures, especially under extreme heat events (such
as the one in August 2021). In a future where a significant fraction of consumed energy will emerge from photovoltaics,
decreases in GHI of 10–20 % could have a significant impact on many human activities that are strongly related with solar
energy production. Also, the AOD variations from climatology led to decrease in UVI up to 53 %, in vitamin-D up to 50 %,



in PAR up to 21 % and in GHI up to 17 %, with implications on health, agriculture and energy. Wildfires are part of the wider
problem of the Mediterranean countries and frequency of summer wildfires is predicted to increase in view of the projected
increasing occurrence of summer heatwaves (Zittis et al., 2022). Our results show that extreme wildland fires such as the one
in August 2021 have far from negligible effects on air quality (e.g., aerosol concentrations, aerosol properties, air pollutants)
and solar radiation effective doses related to human health, ecosystems, and energy (e.g., UV index, vitamin-D, PAR, GHI).
According to recent projections by Ruffault et al. (2020) the frequency of heat-induced fire-weather is expected to increase
in the Mediterranean Basin until 2071–2100 under the RCP 4.5 and RCP 8.5 scenarios, by 14 % and 30 %, respectively. In
combination with extreme drought, extreme wind, and prolonged heatwave conditions in the future, it may well be speculated
that the adverse effects of the projected increased frequency and extent of summer wildfires on vitamin-D and PAR, for
example, will worsen across the Mediterranean countries in the future.

## Appendix A: Measuring instrument description

Pandora uses BlickP algorithm to calculate the total optical depth by estimating a synthetic reference spectrum and cross
sections of $NO_2$ at effective temperature of 254.5 K (Vandaele et al., 1998) are fitted to fourth order polynomial, which results
to the derivation of slant column densities (SCD). Then, it calculates the vertical column densities (VCD) by applying direct
sun air mass factor. In clear sky conditions the precision of the slant column is 0.01 DU (Herman et al., 2009). Measurement
uncertainties related with noise, systematic errors, drift and wavelength shift, are quantified during the monitoring process and
quality flags are provided (Cede and Tiefengraber, 2013). In this study, only the high-quality post processed spectra from the
Pandora actinometer operating at ASNOA are used in order to eliminate any artifacts.

PSR#007 has a global sensor mounted on the auxiliary port and by using the built-in shutter of the instrument, spectral GHI
can be measured. Each cycle of measurements consists of 10 spectra of GHI and 5 dark measurements, that are eliminated and
the average spectra are stored before applying the calibration. Calibrations of the instrument were performed on the field on July
7 and November 3 in 2021 using a 200 W Quartz Halogen lamp that is traceable to Physikalisch-Technische Bundesanstalt
(PTB). The mean ratio between the calibrations was 1.0004 with a range between 0.9902 and 1.0276. Visual inspection of
data showed no possible jump/drift in the time-series. Hence, a linear interpolation between the two calibrations provided the
calibration for each day in the study period (August 2021). The uncertainty budget of the instrument is presented in Gröbner
and Kouremeti (2019), and is less than 1% in VIS, less than 1.7% in UV-A and higher than 2% in UV-B.

The two pyranometers used in this study, manufactured by Eppley Lab., have a black-coated thermopile acting as a sensor (or
detector) which is protected against meteorological conditions by two concentric hemispherical domes. They both comply with
the International Organization for Standardization (ISO) (ISO 9060) criteria for an ISO secondary standard pyranometer, being
classified as "high quality" according to the World Meteorological Organization (WMO) nomenclature (WMO). Additionally,
the corresponding pyranometer measuring the diffuse component (DHI) was mounted on a shading device (Eppley shadow
band) to block the direct irradiance and prevent it from reaching the sensor. Measurements from both pyranometers (for
global and diffuse) have also been corrected for the "dark-signal" offset, also known as "nighttime" offset, which is mainly



due to thermal gradients between the dome and the sensor. As in any optical system that does not use cryogenic cooling or balanced operation, the transfer of infrared radiation between components affects the performance of pyranometers by generating an internal infrared signal that is superimposed to the output signal. The temperatures of the detector and of the outer dome are the main drivers of the temperature gradients that generate the internal, spurious signal. The inner dome acts as a "heat shield"; it reduces the amount of infrared radiation being transferred between the detector and the outer dome (Taylor, 1985). Both pyranometers were calibrated by the Laboratory of Meteorological Device Calibration of NOA (LMDC; Psiloglou, 2021) during 28 and 30 of June, 2021. In order to ensure high-quality measurements, LMDC follows the standard calibration procedure for thermopile pyranometers (ISO 9847), with exposure to real sunlight conditions and comparison with a working standard thermopile pyranometer (Secondary Standard), under constantly clear-sky conditions and for solar altitude greater than 20 degrees. This method is simple and provides sufficient accuracy because errors related to the dependence on solar incident angle and the instrument's spectral response are avoided. Traceability is ensured as LMDC's reference pyranometer, a Kipp & Zonen CMP21 (S/N: 150561), is regularly calibrated in PMOD/WRC, Davos, Switzerland. Also, utilizing the measurements during the nighttime period, from 21:00 to 3:00 of the following day, it was possible to calculate the dark-signal error and correct the measurements of both pyranometers.



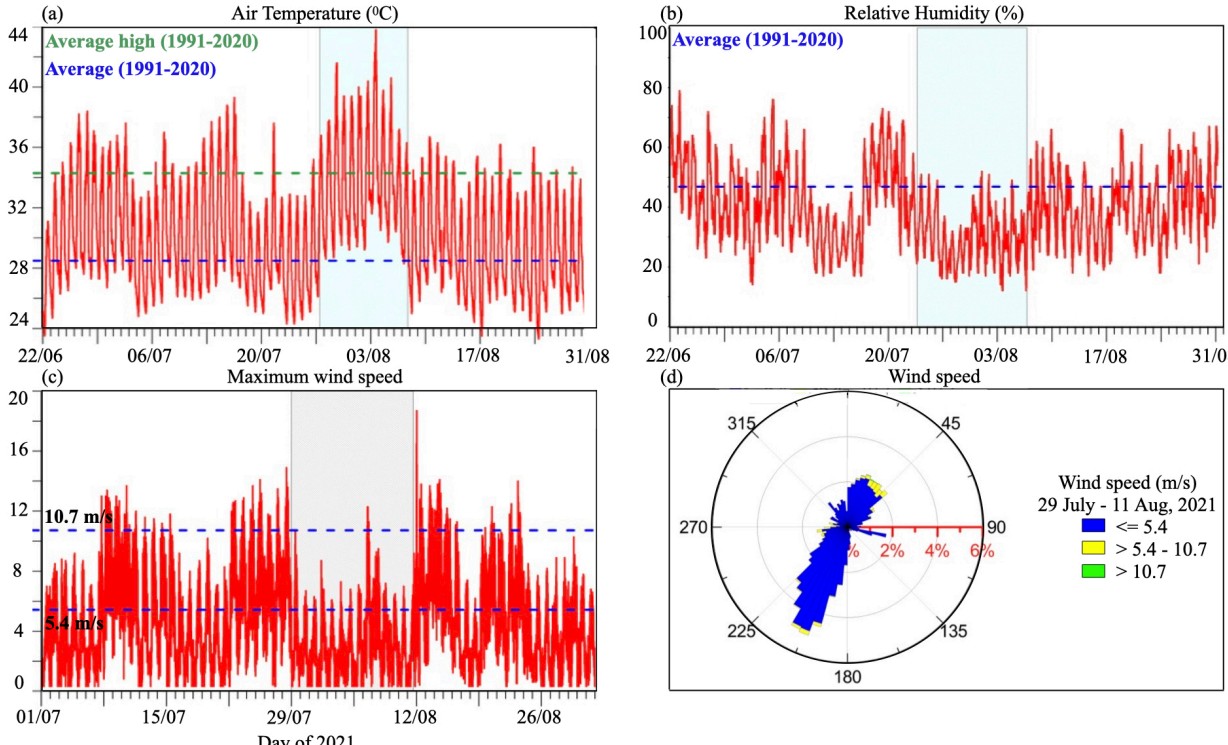

**Figure A1.** Variation of (a) air temperature, (b) relative humidity, (c) maximum wind speed and (d) wind speed during August 2021 in Athens.



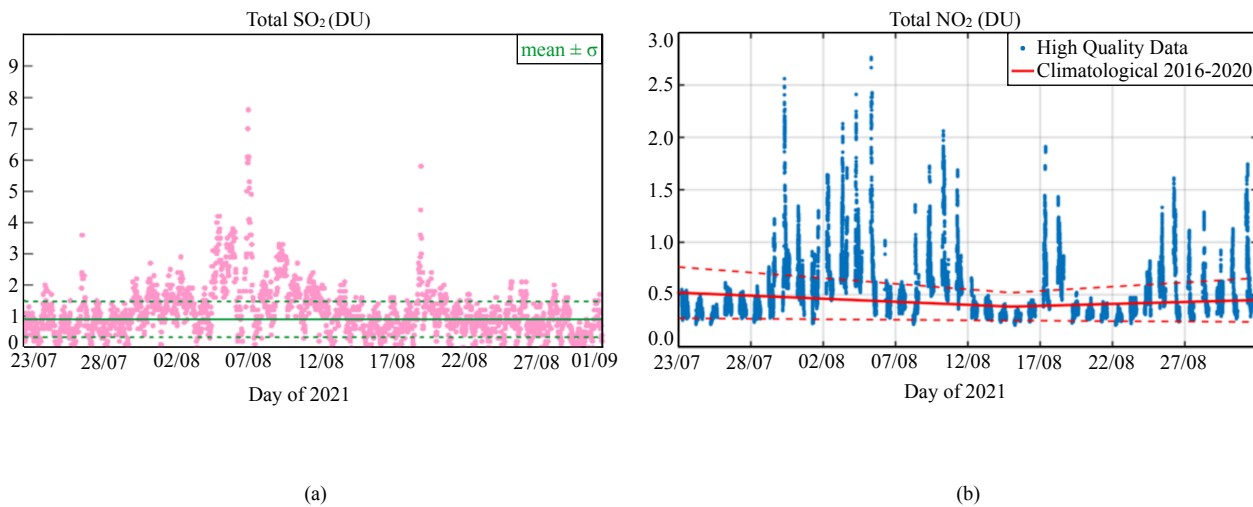

(a)                                               (b)

**Figure A2.** Variation of (a) total $SO_2$ (DU) from Brewer and (b) total $NO_2$ (DU) from Pandora during August 2021 in Athens.



**Table A1.** Daily average values (maximum values in bracket) of aerosol properties for smoke and dust events of August 2021

| Properties | Aug 4 | Aug 5 | Aug 7 | Aug 11 | Aug 18 | Aug 19 | Aug 25 | Aug 26 |
|---|---|---|---|---|---|---|---|---|
| Event | Dust & smoke | Dust & smoke | Smoke | Dust | Smoke | Smoke | Dust | Dust |
| AOD340 | 1.14 (2.16) | 0.93 (2.33) | 1.19 (3.20) | 0.83 (1.37) | 1.03 (3.59) | 0.77 (1.31) | 0.37 (0.58) | 0.40 (0.50) |
| AOD440 | 0.82 (1.57) | 0.75 (2.36) | 0.75 (2.06) | 0.65 (1.17) | 0.74 (2.55) | 0.55 (0.93) | 0.28 (0.42) | 0.34 (0.44) |
| AOD675 | 0.47 (0.78) | 0.53 (1.41) | 0.32 (0.76) | 0.46 (0.94) | 0.34 (1.08) | 0.29 (0.72) | 0.14 (0.21) | 0.26 (0.36) |
| AOD870 | 0.37 (0.54) | 0.40 (0.89) | 0.20 (0.44) | 0.40 (0.87) | 0.21 (0.61) | 0.21 (0.67) | 0.10 (0.15) | 0.23 (0.33) |
| AOD1020 | 0.32 (0.44) | 0.34 (0.68) | 0.15 (0.34) | 0.37 (0.86) | 0.16 (0.42) | 0.18 (0.66) | 0.08 (0.12) | 0.22 (0.31) |
| AE 440-870 | 1.11 (1.56) | 1.82 (1.04) | 1.95 (2.41) | 0.72 (0.89) | 1.73 (2.13) | 1.54 (1.98) | 1.52 (1.67) | 0.56 (0.97) |
| Total AOD500 | 0.68 (1.24) | 0.59 (1.43) | 0.58 (1.53) | 0.57 (1.07) | 0.56 (2.00) | 0.44 (0.84) | 0.22 (0.33) | 0.30 (0.41) |
| Fine AOD500 | 0.43 (0.99) | 0.36 (1.21) | 0.51 (1.49) | 0.26 (0.34) | 0.49 (1.95) | 0.34 (0.70) | 0.18 (0.29) | 0.11 (0.13) |
| Coarse AOD500 | 0.24 (0.28) | 0.23 (0.27) | 0.06 (0.17) | 0.31 (0.74) | 0.06 (0.08) | 0.10 (0.57) | 0.04 (0.08) | 0.19 (0.28) |
| FMF500 | 0.61 (0.80) | 0.53 (0.85) | 0.87 (0.99) | 0.46 (0.54) | 0.84 (0.98) | 0.80 (0.96) | 0.81 (0.92) | 0.38 (0.57) |
| SSA440 | - | 0.87 (0.90) | 0.93 (0.99) | 0.89 (0.90) | 0.94 (0.97) | 0.95 (0.99) | 0.93 (0.97) | 0.89 (0.90) |
| SSA675 | - | 0.94 (0.97) | 0.90 (0.99) | 0.95 (0.97) | 0.94 (0.96) | 0.94 (0.99) | 0.92 (0.97) | 0.95 (0.97) |
| SSA870 | - | 0.96 (0.99) | 0.88 (0.98) | 0.96 (0.98) | 0.94 (0.95) | 0.93 (0.99) | 0.91 (0.96) | 0.96 (0.99) |
| SSA1020 | - | 0.97 (0.99) | 0.86 (0.98) | 0.97 (0.99) | 0.94 (0.95) | 0.92 (0.99) | 0.91 (0.96) | 0.97 (0.99) |

*Author contributions.* AM was the main author of the paper, AM, IF, SK and KE were the main concept organizers and main contributing writing authors. KE organized the ASPIRE campaign that most of the data were collected. IPR, DK, IF and NK has contributed with spectral solar and Pandora measurement analysis, AK and SS with air mass trajectory modeling, KP and IF with radiative transfer modeling and solar radiation analysis, EM, AG and VA have contributed with the Antikythera aerosol data, BP with solar radiation data and analysis, DF with
meteorological data and analysis, VS and AK with Sky camera data analysis, DK and NM with in situ aerosol data provision and analysis, CZ, KE and SK with paper overview and section organization.

*Competing interests.* The contact author has declared that none of the authors has any competing interests.

*Acknowledgements.* Authors would like to acknowledge the Hellenic Foundation for Research and Innovation (H.F.R.I.) under the "First Call for H.F.R.I. Research Projects to support Faculty members and Researchers and the procurement of high-cost research equipment
grant" (Atmospheric parameters affecting Spectral solar IRradiance and solar Energy (ASPIRE), project number 300). AM acknowledges ACTRIS–CH (Aerosol, Clouds and Trace Gases Research Infrastructure–Swiss contribution) funded by the State Secretariat for Education, Research, and Innovation, Switzerland.SK would like to acknowledge the COST Action "Harmonia" (grant no. CA21119), supported by COST (European Cooperation in Science and Technology). This research was supported by the European Research Council (ERC) D-TECT project under the European Community's Horizon 2020 research and innovation framework programme (grant agreement no. 725698), the



ACTRIS preparatory phase project under European Union's Horizon 2020 Coordination and Support Action (grant agreement no. 739530), the PANGEA4CalVal project under the European Union's Horizon Widera 2021 Access program (grant agreement No. 101079201), and the Hellenic Foundation for Research and Innovation (H.F.R.I.) under the "3rd Call for H.F.R.I. Research Projects to support Post–Doctoral Researchers" (Project Number: 7222). This research was also supported by data and services obtained from the PANhellenic Geophysical Observatory of Antikythera (PANGEA) of the National Observatory of Athens (NOA), Greece and by the project "PANhellenic infrastructure

for Atmospheric Composition and climatE change" (MIS 5021516) which is implemented under the Action "Reinforcement of the Research and Innovation Infrastructure", funded by the Op- erational Programme "Competitiveness, Entrepreneurship and Innovation" (NSRF 2014–2020) and co-financed by Greece and the European Union (European Regional Development Fund). NOA team acknowledges the support of Stavros Niarchos Foundation (SNF). We acknowledge Mr. K. Psychas from the Hellenic Ministry of Environment and Energy for providing the air quality measurements for Athens.

*Financial support.* The research work was funded by the Hellenic Foundation for Research and Innovation (H.F.R.I.) under the "First Call for H.F.R.I. Research Projects to support Faculty members and Researchers and the procurement of high-cost research equipment grant" (Atmospheric parameters affecting Spectral solar IRradiance and solar Energy (ASPIRE), project number 300).



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
