# Peer review of "Investigation of the effects of the Greek extreme wildfires of August 2021 on air quality and spectral solar irradiance"

_EGUsphere, 2023_

## Author Comment (AC1)

The manuscript of Masoom et al., "Investigation of the effects of the Greek extreme wildfires of August 2021 on air quality and spectral solar irradiance" presents an analysis of various air quality, aerosol, and radiation measurements during wildfire events in August 2021 in Greece. These wildfire episodes offer interesting cases to study, and indeed nice variety of measurements were utilized. However, there are problems in the current form of the manuscript, which made a thorough and proper evaluation difficult. It seemed that the analysis was not done carefully and thoroughly enough and then an unfinished version of the manuscript was submitted.

We have done major revisions to the text and updated the manuscript based on the comments and suggestions of the reviewer. All figures have been checked and updated, additional aerosol Level 2.0 data from ATHENS-NTUA AERONET station have been used and a new co-author, Prof. Alexandros Papayannis, has been added for the ATHENS-NTUA data. All the changes made in the manuscript are marked in red or are highlighted.

Main comments

**Comment #1:** It was possible to me as a referee (and it will be possible to the readers of the final manuscript) to try to better understand the analysis regarding the AERONET measurements since the data are publicly available. Next, I will mainly clarify the problem points I see, concentrating on the use of AERONET measurements. However, I could not help but get the impression that there might be similar problems with the analysis of other measurements as well (but I cannot access them equally readily); the problem being that the analysis did not seem always careful enough and the justification of the obtained results always physically reasonable.

**Response#1**: We made appropriate corrections and now explain better the analysis regarding the AERONET measurements and provide more information for the used AERONET data (see also Response#3, Response#7, and other responses). As for the other measurements of the other instruments used, we checked and there were not similar problems, as the reviewer had the impression. We now provide more information relative to the calibration of the instruments and the quality assurance of the data in the revised Appendix A. Section 2.1.2 regarding the analysis of the AERONET data has been fully revised and now reads as follows:

"*ATHENS–NOA AERONET (Aerosol Robotic Network) station was operating from 2008 to 2021, with a CE318 sun/sky–photometer from Cimel Electronique (CIMEL#440) in operation during the study period. The columnar aerosol optical depth (AOD), Ångström exponent (AE), fine/coarse AOD, single scattering albedo (SSA) and volume size distribution (VSD) (Dubovik and King, 2000; Dubovik et al., 2006; Sinyuk et al., 2007), retrieved from AERONET Version 3 algorithm (Giles et al., 2019) are used here. For Athens, Level 2.0 AERONET direct sun products (AOD, AE, fine/coarse AOD) were used in this study except for the days with very high smoke and/or dust aerosol load (August 4, August 5, August 7, August 11, August 18 and August 19), when Level 1.0 data was used. For these particular days, the AERONET automatic cloud screening algorithm filtered out data related with the wildfire plumes, when going from Level 1.0 (unscreened) to Level 1.5 (cloud-screened) and Level 2.0 (cloud screened and quality assured) products, due to the very high temporal variations of the AOD. The sun photometer measurements during high aerosol events with extremely frequent changes of the radiation field are difficult to be captured due to cloud flagging algorithm failure, and are more likely to be rejected as cloudy, even in cloud-free situations (Evan et al., 2022). Manual control of sky-camera (SKYCAM) images from the cloud camera was used as additional evidence for non-cloud presence, on the choice of the Level 1.0 products to be used (Appendix Figure A1). Accordingly, Level 2.0 inversion products (SSA, VSD) were used except for the days mentioned above (where Level*

*1.5 data with sky-error limit up to 5% was used with additional filtering of solar zenith angle (SZA) > 45° and coincident AOD at 440 nm > 0.4 for SSA) since the strict criteria for Level 2.0 filters out a lot of useful retrievals in summer months, as explained in Kazadzis et al. (2016). The approach of using lower–level data increases the uncertainty of the retrievals, but the evidence by the collocated data of other sources provides a relatively high degree of data quality assurance. Also, the climatological values of the aforementioned properties reported in previous studies (Raptis et al., 2020) are used as reference. We note here that measurements from the NTUA AERONET station operating in Athens since January 2021 (CE318 sun/sky-photometer) were used for August 7, when the data in morning hours was not available from the ATHENS-NOA AERONET station. The same parameters (AOD, AE, Fine/Coarse AOD and SSA) were also collected in PANGEA observatory. For PANGEA, Level 2.0 products were used for both direct sun and inversion products, as there was not much difference in Level 1.0 and higher-level products, as was the case of Athens."*

**Comment #2:** Based on that part of my evaluation, which concentrated on AERONET analysis (and MODIS too, as explained below), it seemed that in your analysis not enough attention was paid to the quality of the measurements and furthermore they were not included and analyzed in a fully consistent way. Moreover, the justifications for your findings were not always clear or convincing. With the related points, which I explain below, I will not try to make an exhaustive list; they are just a few examples. I think the authors need to do a major revision and carefully check and improve the entire manuscript.

**Response#2:** We thank the reviewer for the comment. We have put special attention on the quality of the measurements. In the case of AERONET, with the help of sky camera images (some of which are provided in Appendix Fig. A1) we were able to find cases for which the AERONET cloud flagging algorithm flagged the data while the variability was actually not due to clouds but because of smoke.

The use of AERONET Level 1.0 data was made only for the wildfire days (for remaining days. L2.0 data was used) as we have explained in Response#1 and have been accompanied by manual inspection of cloud camera pictures. We believe that manual inspection of images is one of the most accurate ways to approach this issue. To be specific, we present the cases (4, 5, 7, 11, 18, 19 August) for which we selected to use L1.0 instead of L2.0 data.

On August 7 for instance and focusing on the time period between 10 UTC and 14 UTC, most of the data is flagged as cloudy in Level 1.5 and Level 2.0 (Level 1.5 data from Athens capture only 3 measurement points). But in the sky camera images, it is clearly visible that these pixels are cloudless and there is smoke (Figure A1 c3-c5). Hence, we have used Level 1.0 data (15 measurement points) because most of the information was lost in Level 1.5 even though these were cloudless cases. However, in the earlier version of the manuscript, we did not provide an insight into this. Hence, we have attached some images of sky-camera to justify the selection of Level 1.0 data (Figure i).

[Figure]

Figure A1: Sky-camera images for Athens (a1-a5), (b1-b5), (c1-c5), (d1-d5), (e1-e5), and (f1-f5) for August 04, 05, 07, 11, 18 and 19, respectively.

Specific comments

**Comment #3:** The discussion in the page 20 was one of the most unclear and unconvincing ones. Did I follow correctly that you compared Antikythera and Athens Inversion data in the very same day and so that Antikythera measurements were mostly carried earlier in time than those in Athens, but you nevertheless explained that the comparison gives information about aerosol aging during the

transport? Is this a correct interpretation? If I compare Level 1.5 of that day, I see that Antikythera and Athens measurements starts from 04:58 UTC, 14:20 UTC, respectively.

**Response#3**: The unconvincing discussion on page 20 has been fully revised. As the reviewer commented, indeed there were mis-sights in our analysis relative to the transfer of smoke from Athens to Antikythera. It is very possible – as the reviewer indicated - that smoke over Antikythera on August 7[th] was not transferred solely from Attica. Thus, we removed the discussion relative to the transfer and the ageing of smoke and we discuss aerosol optical properties over Antikythera independently from the aerosol optical properties of the smoke from the Attica wildfires in section 3.5.

We now discuss that fires at Peloponnese on 6[th] of August could have an effect on Antikythera. Even if air mass trajectories are clearly bringing air masses from the North, remaining smoke of the previous day could have an effect on measurements during the whole day, the 7[th] of August. Smoke from Athens to Antikythera arrived in mid-day of August 7[th] according to the trajectory analysis and the Lidar signals. L2 AERONET measurements at Antikythera (revised Figure 9 in the updated manuscript, also included here), in the afternoon hours only, show higher SSAs than Athens. We now mention in the revised section 3.5: "*Higher SSA values in the afternoon of August 7 at PANGEA, compared to Athens, could be an indication of changing optical properties of smoke through transport and ageing processes that reduced the absorbing capability (Dasari et al., 2019). However, the presence of smoke from Peloponnese local fire makes this assumption quite uncertain.*"

[Figure]

Figure 9. Single scattering albedo for (a) Athens (AERONET Level 1.5 with filters as mentioned in Section 2.1.2) and (b) PANGEA (only afternoon measurements from AERONET Level 2.0 inversion) for August 07, 2021.

For PANGEA, Level 2.0 products were used as there was not much difference in Level 1.0 and higher-level products as was the case with Athens.

**Comment #4:** After the line #435, the explanations are then simply not possible to follow. SSA is an extrinsic property, so some more explanations would be needed for your discussion regarding dilution during the transport. And even if the dilution would have been in principle a physically reasonable argument, AOD levels in Antikythera on that day (August 7th) are much higher than in Athens? Did you consider it and what was your interpretation, that during the transport AOD increased substantially? I think there was entirely other source of wildfire affecting Antikythera (as I explain below). The unclear reasoning continues in that paragraph: what density would have to do with SSA spectral shape, etc?

**Response#4**: We agree with the reviewer, and we have modified this part (see also Response#3) and the revised Section 3.5 which reads now as follows:

*"On August 7, smoke was detected from the Polly[XT]–NOA lidar above PANGEA observatory at altitudes 0.5–3.0 km above the surface. Fig. 7 shows the Polly[XT]–NOA lidar attenuated backscatter coefficient at 1064 nm and the 3–day air masses back trajectories above the station (ending at 12:00 UTC on August 7, 2021) from FLEXPART–WRF and HYSPLIT model simulations (Fig. 7b and 7c). Wildfire aerosol sources and transports, lidar measurements and analyses with different models confirm that the smoke plume was transferred to PANGEA from various fire events, as can be seen from the layers below 3 km in Fig. 7a.*

*In the morning of August 7, aerosol concentrations above PANGEA were increased at different heights, probably as a result of remaining smoke plumes of a fire that started and also ended on August 6 in southern Peloponnese (Mani peninsula), an area close to the island of Antikythera (PANGEA). Air mass trajectories also showed that smoke from the main Attica and Evia fires arrived in PANGEA (at altitude below ~1500 m) in the midday of August 7 (Fig. 7a – area 4), possibly mixed with smoke from wildfires that were burning in Peloponnese (at altitude above ~2000 m) during the previous day and/or night. The fire in Mani peninsula on August 6 can be seen in Aqua MODIS satellite images (see for instance Aqua MODIS corrected reflectance, NASA WorldView; https://go.nasa.gov/3SEK9XK (MODIS)).*

*Figure 8 presents the diurnal variation in total, fine- and coarse-mode AODs at 500 nm and AEs on August 7, 2021. In Athens, high AOD values (up to ~0.75) were observed in the early morning hours that were further increased after 10 UTC (up to ~1.53), which were accompanied by high fine-mode fraction. The AE for both stations was found to be above 1 for the entire day. The lower AE values at 340-440 nm compared to those at 500-870 nm indicate a negative curvature effect, signifying the dominance of fine particles (Schuster et al., 2006). At PANGEA, the AOD was high in the morning (~0.90) and afternoon (~0.72) hours of August 7 (due to the Athens fire transport), with significantly high fine-mode AOD (~0.85 and ~0.68, respectively) and high fine-mode fraction. The fires of southern Peloponnese may have also affected the air composition on August 7 at PANGEA.*

*Figure 9 shows the daily-averaged spectral SSA variations from 440 nm to 1020 nm on August 7, when smoke was present over Athens and over PANGEA (only afternoon values). The median SSA value at PANGEA decreases monotonically from 0.95 at 440 nm to 0.90 at 1020 nm. In Athens, the median SSA value was found to have a more drastic decrease from 0.91 at 440 nm to 0.82 at 1020 nm. It should be noted that Level 1.5 inversions were used for SSA retrievals in Athens from ATHENS-NOA station and one morning measurement was taken from ATHENS-NTUA to get the average SSA of the day. The decreasing SSA with wavelength indicates the presence of fine smoke, which is evident for both stations. Higher SSA values in the afternoon of August 7 at PANGEA, compared to Athens, could be an indication of changing optical properties of smoke through transport and ageing processes that reduced the absorbing capability (Dasari et al., 2019). However, the presence of smoke from Peloponnese local fire makes this assumption quite uncertain."*

**Comment #5:** I would argue that the whole comparison between the two stations on that day is not meaningful, if the "end station" measurements are made before the "starting station". However, I would argue that there is even a much stronger point, wildfire event that strongly affected Antikythera and not Athens, which you did not discuss. And this is the reason daily mean AOD_500 in Antikythera and Athens (on August 7th) were 0.53 and 0.38, respectively. This fire was in Mania peninsula and can be seen in the Aqua of the day before (hopefully the link becomes ok, but if not,

please do see the Aqua MODIS corrected reflectance from August 6th, for instance from NASA WorldView). https://go.nasa.gov/3SEK9XK

**Response#5**: The reviewer is right, and the manuscript has been modified according to the comment (see also Response#3 and Response#4).

We agree and we have mentioned the mix of the Mani and Athens fire for the 7th of August and the overall discussion about not being sure on SSA changes due to transport and ageing.

**Comment #6:** This seems to have been a quite strong and active fire and estimating the prevailing wind directions, "interpolating" from your trajectories, this fire has very likely clearly affected Antikythera measurements in the morning of August 7th. Do you disagree?

**Response#6**: We thank the reviewer for the comment. In principle, we agree and changed the manuscript properly. Inspection of the Aqua MODIS corrected reflectance images for August 6th, kindly provided in reviewer's comment#6 (https://go.nasa.gov/3SEK9XK) shows that the smoke from Mani fire did not pass explicitly over the small island of Antikythera during the day of the 6th but during the night and most probably some remaining smoke of this fire (extinguished at the night of the 6th) affected Antikythera on the 7th. On the 7th afternoon most of the existing smoke captured from Lidar, AERONET was from Athens as the trajectories show clearly. In general, we think that Athens smoke prevailed at Antikythera on the 7th afternoon, but as mentioned, arguments about change of absorption during transport are uncertain.

**Comment #7:** Regarding AERONET measurements, you decided to use Level 1.0 Direct Sun product and Level 1.5 Inversion product. One can justifiably say that this should not be done. But at the very least, then the measurements should have been studied more closely (what to include in the analysis and what not). This did not seem to have happened.

**Response#7**: For the presentation of Athens data (direct sun) in the revised figure 4 we have used L2 data for all days with the exception of days August 04, August 05, August 07, August 11, August 18 and August 19, for which we used L1.0 data and also sky camera pictures in order to visually identify the presence of clouds. For the inversion products (figure 4) we have used L1.5 products with sky-error limit up to 5%, solar zenith angle > 45° and coincident AOD at 440 nm > 0.4 (new plots).

For the Athens analysis (revised figure 8) for August 7th we have used direct sun L1.0 data and sky camera proof (visual inspection was possible for the limited number of days for which data was analyzed) and L1.5 inversion data with sky-error limit up to 5%, solar zenith angle > 45° and coincident AOD at 440 nm > 0.4 (new plot). For Antikythera, we have used L2.0 for both direct sun and inversion products (SSA) in revised figure 8.

This explanation has been added in the revised section 2.1.2 and also described in Response#1.

**Comment #8:** There is a set of criteria used to decide whether a measurement in Inversion product will rise from Level 1.5 to Level 2.0 (limit in sky-error, SZA, in AOD at 440nm etc). I would argue that those should have been considered measurement-by-measurement even in the case if Level 1.5 measurements are used (to include some quality assurance). However, since this was not done, some results are reported that might not have been that reasonable to emphasize at all. For instance, in line #366 ("SSA reaches very low values (even below 0.7)"), this particular SSA measurement, you refer to, is from August 12th when during the measurement AOD at 440nm was 0.15 and the sky-error 8.5%. I do not think it is meaningful to report and discuss AERONET SSA (highly uncertain) measurement during this kind of low-AOD conditions?

**Response#8**: We agree with the reviewer and we modified the analysis as explained in Comment#7.

**Comment #9:** Line #354: "maximum AOD values between 0.78 and 0.39 at C5". This was a relatively clean day, while only the two last measurements of the day (in Level 1.0) show a drastic increase, and this is exactly from where you draw your conclusions of maximum AOD values? Do you really trust in these measurements? If you plot AE, you see that it drops very quickly at the very same time (and the coarse mode AOD suddenly rises accordingly). Maybe this was not a very clear smoke day at all (quite low AOD during entire day and then two cloud contaminated AOD measurements in the end of the day)? At least I would suspect that these two measurements are clearly cloud contaminated ones (sudden AOD_coarse (AE) increase (decrease)), this is why AOD increases rapidly. Overall, the mean daily AOD_500 ~ 0.11 would suggest that his was a quite clean day. I was left with an impression that you did not analyze AERONET measurements very carefully.

**Response#9:** We thank the reviewer. We repeated the analysis using L2 data for this day (August 17) and the points have been eliminated. This day is not discussed any more, text and figure 5 have been revised.

**Comment #10:** Also related to this comparison of two stations on August 7th, did you really use consistently Level 1.5 inversion product? It seems to me that your Athens data was based on Level 2.0 and Antikythera on Level 1.5 (if both were from Level 1.5 the spectral difference seems much smaller) on August 7th – is this correct interpretation. But if you did not use the AERONET measurements consistently, why was that?

**Response#10:** We clarify that for Antikythera we used L2 direct sun products, for Athens L1.0 products with camera pictures (revised figure 8) and for inversion products, L1.5 data with quality criteria were applied for Athens and L2.0 for PANGEA (revised Figure 9). Please also see our previous responses.

**Comment #11:** From time to time the manuscript started "suddenly" with a discussion that was difficult to really link with the discussed just before. Just one example: how the discussion starting at the line #367 is explaining the results introduced just earlier in the paragraph? It was not possible to follow.

**Response#11:** We have improved the manuscript and made it more consistent. The line has been revised and now reads as follows "*the lower SSA values in this day indicate the presence of strong absorbing aerosols (Kaskaoutis et al., 2021; Wu et al. (2021)).*"

Please see Case 1 of Section 3.3.

**Comment #12:** Line #396, "High AOD and high AE in morning and high AOD and low AE in evening". Was it really so? If I look at the diurnal pattern, is see that AOD is drastically lower and quite low in the evening than in the morning. Do you agree? This is just one example of sentence or statement that left the impression that the work was not done thoroughly and carefully enough.

**Response#12:** We agree with the reviewer and have corrected this sentence accordingly.

"*For that particular day, a constant dust layer and a decreasing from morning to afternoon smoke layer led to a decrease in AOD and in AE during the day*".

**Comment #13:** Line #441, what is PM there?

**Response#13:** It has been removed while updating the earlier discussion.

**Comment #14:** Line #489: with the sentence, starting here, you explain why AE is larger in smoke than dust dominated case. However, it does not explain why AE would affect GHI change as you found, which the reader assumed to read about (and what would be interesting and relevant, compared to the pretty obvious explanation you gave, which was not relevant in that context).

**Response#14:** It has changed as follows:

*"It is interesting that although the daily average AOD at 500 nm is almost same, ~0.58, the average AE at 440-870 nm is 1.97 and 0.74 on August 7 (smoke) and August 11 (dust), respectively, signifying the large variation in AOD with wavelength on August 7 (1.18 to 0.15 for the 340-1020 nm) compared to that on August 11 (0.80 to 0.35 for the 340-1020 nm) (Table A1 Appendix). Hence, the change in the composition of GHI is significantly more pronounced on August 7 and the attenuation is more enhanced mainly because of stronger absorption of light by smoke aerosols relative to dust aerosols (Kaskaoutis et al., 2021)."*

**Comment #15:** Figure 10: perhaps not the best legend titles used in the upper plots (not all the simulations are at UV wavelengths).

**Response#15:** The legend of Figure 10 has been corrected.

**Comment #16:** Comment: perhaps interesting additional information would have been available if you studied the wavelength dependence in AE (so called spectral curvature effect) or AAE during these episodes.

**Response#16:** It has been added (as shown in the figure below) to Figure 8 in the manuscript.

[Figure]

Figure 8: Variation of total AOD, fine-mode AOD and coarse-mode AOD at 500 nm, and Ångström exponents in (a) Athens (AERONET Level 1.0) and (b) PANGEA (AERONET Level 2.0) during the wildfire event of August 7, 2021.

Text added in section 3.5:

*"The AE for both stations was found to be above 1 for the entire day. The lower AE values at 340-440 nm compared to those at 500-870 nm indicate a negative curvature effect, signifying the dominance of fine particles (Schuster et al., 2006)."*

We thank the reviewer for the valuable comments that helped us improve the quality of this manuscript.

**Reference**

[revised manuscript text omitted]

---

## Author Comment (AC2)

**Response to Reviewer 2**

We have done major revisions to the text and updated the manuscript based on the comments and suggestions of the reviewer. All figures have been checked and updated, additional aerosol Level 2.0 data from ATHENS-NTUA AERONET station have been used and a new coauthor, Prof. Alexandros Papayannis, has been added for the ATHENS-NTUA data. All the changes made in the manuscript are marked in red or are highlighted.

General comments:

**Comment #1:** The article provides a description of the analysis performed during wildfire events occurred in August 2021 in Greece, using data from several active and passive remote sensing instruments combined with modelling data. The objectives of this study cover different issues related to wildfires regarding the air quality, the attenuation of surface levels of solar UV and global radiation, and the spatial and temporal aerosol spectral optical properties. Although the paper is adequately structured, the methodology is quite properly defined, and the analysis is rather comprehensive, however in some parts of results the reading is not fluent making difficult the comprehension of this section. Therefore, before publication there are some issues which the authors should take into consideration.

The main issue is related to the calibration of instruments used in this study. This particular point merits a serious revision. Specific comments are reported below.

**Response#1**: We have revised the manuscript and figures and made the text in the sections fluent and comprehensive. We have provided the information relative to the calibration of the instruments and the quality control of the data in the revised Appendix A.

**Specific comments**

Comment #2: L40: more clarity is required in "the spectral response also depends..."

**Response#2**: We have restructured this sentence for better understanding as

"In another study by Park et al. (2018), it was found that smoke reduces significantly the ultraviolet (UV) actinic flux, while the spectral light absorption by smoke aerosols depends upon the combustion conditions, smoke chemical properties and atmospheric processing (Saleh et al., 2013; Srinivas et al., 2016)."

(Please see Section 1, paragraph 2)

**Comment #3:** L90: The purpose of ASPIRE campaign could be specified as being independent on wildfire events in 2021.

**Response#3**: We have added the purpose of ASPIRE campaign as

"The ASPIRE campaign was designed with the objective to investigate the effect of clouds, aerosols, water vapour and absorbing trace gases on spectral solar irradiance and contributes to interdisciplinary aspects. Wildfire events during this campaign allowed the in-depth investigation of atmospheric composition and its impact on the transfer of solar radiation using active and passive remote sensing instruments."

Please refer to Section 1, paragraph 6).

**Comment #4:** L110: The authors should explain the meaning of actinometric station for readers not familiar with it.

**Response#4**: We mean a station with instruments for the observation and measurement of solar radiation. A brief description of actinometric station is provided as follows in Section 2.1.

"In addition to the instruments that are permanently installed and operating at the actinometric station (for observations and measurements of solar radiation) of NOA (ASNOA)....."

**Comment #5:** L115: More details should be provided before Table 1 since the acronyms (such as PSR, GNAPMN ...) are not specified for most of instruments. Yet, the same is for the column "quantity" (VSD, etc ..). In addition, in column "description" the authors should provide the same type of information. Infact in this column" the time sampling of data is reported for some parameters, for others the wavelength or wavelength ranges. The authors should give the same type of information (if possible) for all the parameters. An addition column with a reference can be included.

**Response#5**: We have provided the description of the acronyms GNAPMN as Greek National Air Pollution Monitoring Network}, GAA as Greater Athens Area, ASNOA as actinometric station of NOA, BRFAA as Biomedical Research Foundation of the Academy of Athens, NOA as National Observatory of Athens and PANGEA as PANhellenic GEophysical observatory of Antikythera in table footnote. Also, we have expanded terms like AOD, SSA, VSD, PSR in the table. For column 4, it is not possible to provide same type of information for all the parameters as different instruments used here measure different quantity. Hence, we have renamed this column from "Description" to "Temporal/Spectral Resolution". (Please refer to the updated Table 1)

**Comment #6:** Erythemal irradiance and Viatmin D dose are cited without being explained. In column"Type", Vaisala is the name of the company.

**Response#6**: Has been corrected to "Vaisala CL31" in updated Table 1.

**Comment #7:** L125: no information of Brewer calibration for ozone and UV irradiance is reported.

**Response#7**: Information about the Brewer calibration is provided in the revised Appendix A as follows:

"Brewer#001 is measuring automatically the direct, diffuse and global spectral irradiances in the UV and visible regions since 2003 and every two-three years it is calibrated on site by International Ozone Services (https://www.io3.ca/). Since 2020, the Brewer is calibrated using a set of three 200-Watt lamps that are traceable to the scale of spectral irradiance established by the Physikalisch-Technische Bundesanstalt (PTB). More detailed information about the Brewer including measurements, quality control/assurance procedures, and calibration can be found in (Eleftheratos et al., 2021; Diémoz et al., 2016). The uncertainty in the Brewer measurements is estimated to 5 % for wavelengths above 305 nm and SZAs lower than 70° (Garane et al., 2006). There is about 1 DU uncertainty in Brewer direct sun SO2 measurements (Fioletov et al., 1998). During the wildfires, the SO2 levels rose high enough in Athens, well above the mean  $\pm 2\sigma$  (with mean being 0.9 DU and  $\sigma$  being 0.6 DU), and hence, the uncertainty was not of much importance as it is in the case of low SO2 values."

**Comment #8:** SO2 measured by Brewer is affected by large uncertainty. How did the authors manage this kind of observations?

**Response#8**: There is about 1 DU uncertainty in Brewer direct sun SO2 measurements (Fioletov et al., 1998). During the wildfires, the SO2 levels rose high enough in Athens, well above the mean  $\pm 2\sigma$  (with mean being 0.9 DU and  $\sigma$  being 0.6 DU), and hence, the uncertainty was not of much

importance as it is in the case of low SO2 values. This description has been added in the revised Appendix A (also mentioned here in Response#7).

**Comment #9:** L190: The Erythemal reference action spectrum that should be used and acknowledged is ISO/CIE 17166-2019.

It is not clear which pre-vitamin D3 action spectrum was used to determine vitamin dose Is that reported in R. Bouillon, et al., (Action spectrum for production of pre-vitamin D3 in human skin, CIE Technical Report 174, 2006)?

Why was calculated the dose for the vitamin D whereas for erythema the dose rate?

**Response#9**: We have added the recommended references (ISO/CIE and Bouillon et al., 2006) as well as the following description (about dose and dose rate) in Section 2.1.4, paragraph 3.

"For vitamin D and PAR, we are interested in the cumulative daily dose (since their effects depend on the overall dose that a human or a plant, respectively, gets), while for erythema, we are interested in dose rates around the local noon, when solar radiation is higher."

**Comment #10:** L251: The geographical coordinates of the locations can be included.

**Response#10**: We have added the coordinates of the location in Figure 1.

**Comment #11:** L257 Figure 1 it is not clear the usefulness of sky-camera images without any explanations as these images are related to time which is different from that of Figure 2 (MSG satellite images).

Response: Here, we have used the images when it is indicative of the events and that is why the time is not the same as the time of MERRA-2 images of Figure 2 (these are averages from 05-14 UTC). However, we have changed the 2nd row of column b and c from August 05 images to August 11 images to match with the dates of MERRA-2 images in Figure 2.

**Comment #12:** L258: the variability range (1 std) of the climatic series can be added to Fig. A1 which reports only the averages.

**Response#12**: The variability range has been added in the above-mentioned figure (which is now Fig. A2 in the updated manuscript).

**Comment #13:** L286: in Section 3.2 a table with the stations involved in this analysis could make easier the interpretation of figure 3.

**Response#13**: We have provided a tabular description of the station and the measured quantities in Figure 3 for better understanding.

**Minor remarks:**

Comment #14: All the acronyms should be clarified in the abstract.

Response#14: Done

Comment #15: L55: add the geographical coordinates of Monte Curcio

Response#15: Done

Comment #16: L56: add the year of Spanish wildfires

**Response#16: Done**

Comment #17: Table 1 Viatmin—>Vitamin

Response: It has been corrected.

**Comment #18:** Figure 1(b) the acronym of MSG should be provided. **Response#18:** Done

Comment #19: L278 Figure 2d e and dà Figure 2d e and f

Response#19: Done

We thank the reviewer for the valuable comments that helped us improve the quality of this manuscript.

[revised manuscript text omitted]

---

## Author Response (AR2)

Response to Editor's and Reviewer's Comments

**Comment#1:** Please address the additional comment by Referee #1

Anonymous referee #1: "In my opinion the manuscript has improved considerably. Also, my comments and suggestions are also sufficiently addressed in the revised version. I have only one and very minor comment to add. I think it would be useful info to mention after the sentence (which starts from the line #145 in the track-changed version) that these criteria are same (or similar since SZA limit is not exactly the same) than what is used in AERONET in their quality assurance step from Level15 to Level2. To clarify this reasoning in your effort to ensure that you are using as quality assured data as possible.

**Response#1:** We thank the reviewer for the suggestion. We have updated this sentence accordingly as

"Level 2.0 inversion products (SSA, VSD) were used except for the days mentioned above (where Level 1.5 data with sky-error limit up to 5% and solar zenith angle (SZA) > 45° was used with additional filtering of coincident AOD at 440 nm > 0.4 for SSA *similar to the AERONET quality assurance criterion for Level 1.5 to rise to Level 2.0, except for SZA for which we have used 45° instead of 50° (Holben et al., 2023))."*

**Comment#2:** It is not clear to my why you include data from two trajectory models (HYSPLIT and FLEXPART) and two aerosol reanalyses (CAMS and MERRA-2). For instance, the current use of CAMS for aerosol speciation at Athens and of MERRA-2 for dust transport seems to be quite arbitrary. I suggest to decide on one each to increase consistency of your analysis as a unified use of data sources would give the study a more focussed narrative.

**Response#2:** We thank the Editor for the suggestion. Hence, we now use results from HYSPLIT trajectory model and CAMS reanalysis. Figures showing FLEXPART (Fig. 7b) and MERRA-2 (Fig. 2 d, e, f) analyses are now replaced with HYSPLIT (new Figure 7b in updated manuscript) and CAMS analyses (new Figure 2 d, e, f in updated manuscript), respectively (Please also see attached Figures below). The text has been corrected accordingly and the corresponding references have been changed/deleted.

[Figure]

**Figure. 2.** Identification of dust transfer to Athens using HYSPLIT back trajectories ending at 12:00 UTC (a, b, c) and Copernicus Atmospheric Monitoring Service (CAMS) reanalysis time-averaged dust optical thickness over 00-21 UTC (d, e, f).

[Figure]

**Figure 7.** (a) Lidar attenuated backscatter coefficient at 1064 nm and (b) HYSPLIT 24 h backward trajectories ending at 12:00 UTC on August 7, 2021 over PANGEA.

**Comment#3:** Isn't there a better way to highlight different features in Figures 5 and 7? Circles are fine for presentations. In a paper, you might want to be a bit more quantitative, though.

**Response#3:** We thank the Editor for the suggestion and have removed the circles from Figure 5 and 7 (Please see Response#2 for Figure 7 and Figure 5 is attached below) and replaced them with the probable description of the highlighted layers in these plots.

[Figure]

**Figure 5.** Time-height distribution (a, b, c, d, e, f) of ceilometer attenuated backscatter coefficient at Athens for 6 days of August, 2021 listed in Table 2.

**Comment#4:** While Table 1 has been much improved, it still doesn't seem to be consistent, particularly when it comes to the columns instrument/network and type. There is in fact a mixture of instruments, networks, manufacturer, and type of instrument. For instance, AOD is measured with a sun photometer (instrument) that is part of AERONET (network), manufactured by CIMEL (manufacturer), and measures spectral radiances. In the same way, the backscatter coefficient is measured with a lidar (instrument) of type PollyXT that is part of PollyNet (network).

**Response#4:** Thank you for the comment. The Table 1 has been corrected as recommended, and now reads as follows.

**Table 1.** Description of ground-based measurements

| Quantity | Instrument Name/Network | Location | Temporal/Spectral Resolution | Type | Reference |
|---|---|---|---|---|---|
| PM10, PM2.5 | Eberline, Thermo Fischer/GNAPMN* | GAA* | Daily | Beta gauge particulate monitor | Grivas et al., (2008) |
| NO, NO₂ | Horiba Ltd./GNAPMN | GAA | Hourly | Analyzer APNA | Grivas et al., (2008) |
| Scattering Coefficient | TSI/NOA Network | ASNOA* | 3 wavelengths (450, 550, 700 nm) | Nephelometer | Kaskaoutis et al., (2021) |
| Absorption Coefficient | Magee/NOA Network | ASNOA | 1 min/7 wavelengths (370, 470, 520, 590, 660, 880, 950 nm) | Aethalometer | Drinovec et al., (2015), Liakakou et al., (2020) |
| Columnar NO₂ | Pandora/Pandonia | ASNOA | 10-20 min | Spectral radiometer | Herman et al., (2009) |
| Columnar SO₂ | Brewer/EUBrewnet | BRFAA | - | Spectrophotometer | Kerr et al., (2010) |
| Aerosol optical depth (AOD), Ångström Exponent | CIMEL/AERONET | NOA*, PANGEA* | 15 min | Sunphotometer | Giles et al., (2019) |
| Fine/Coarse AOD | CIMEL/AERONET | NOA, PANGEA | 500 nm | Sunphotometer | O'Neill et al., (2003) |
| Single scattering albedo | CIMEL/AERONET | NOA, NTUA*, PANGEA | 440, 675, 870, 1020 nm | Sunphotometer | Dubovik and King (2000) |
| Volume size distribution | CIMEL/AERONET | NOA | - | Sunphotometer | Dubovik and King (2000) |
| Backscatter coefficient | Vaisala/E-Profile Polly-XT/EARLINET | ASNOA PANGEA | 910 nm 1064 nm | Ceilometer Lidar | Kotthaus et al., (2016) Engelmann et al., (2016), Baars et al., (2016) |
| Spectral irradiance | Precision Spectro-Radiometer (PSR)/NOA Network | ASNOA | 300–1020 nm | Spectral radiometer (SP) | Gröbner and Kouremeti (2019) |
| UV-B irradiance | Brewer/EUBrewnet | BRFAA | 290–315 nm | Spectrophotometer | Garane et al. (2006) |
| Global horizontal irradiance, Diffuse horizontal irradiance | Kipp and Zonen/NOA Network | ASNOA | 285–2800 nm | Pyranometer | WMO (2021) |
| Near Infrared irradiance | PSR/NOA Network Kipp and Zonen/NOA Network | ASNOA | 700–3000 nm | SP Pyranometer | Gröbner and Kouremeti (2019) |
| Erythemal irradiance | Brewer/EUBrewet, PSR/NOA Network | BRFAA, ASNOA | 300–400 nm | Spectrophotometer, SP | Kerr (2010), Gröbner and Kouremeti (2019) |
| Vitamin D dose | Brewer/EUBrewnet PSR/NOA Network | BRFAA, ASNOA | 300–400 nm | Spectrophotometer, SP | Kerr (2010), Gröbner and Kouremeti (2019) |

Additional corrections to the manuscript:
Our figures have been checked according to the Remarks from the preceding review file validation, and colour schemes have been revised accordingly.
- Fig. 1a: Colours revised.
- Fig. 3: Figure enlarged.
- Fig. 4: Line patterns and colours revised.
- Fig. 8: Line patterns revised.
- Fig 11d: Bar, rhombus and line colours revised.

We thank the reviewers and the Editor for the valuable comments and suggestions that improved the quality of the manuscript even further.

[revised manuscript text omitted]

---

## Author Response (AR3)

**Remarks from the preceding review file validation**

1. With the next revision, please include the section "Correspondence to:" in to the title page of the *.pdf manuscript. Please use our templates: https://www.atmospheric-chemistry-and-physics.net/submission.html#templates / Technical instructions for MS Word and compatible formats /. 2. Please ensure that the colour schemes used in your maps and charts allow readers with colour vision deficiencies to correctly interpret your findings. Please check your figures using the Coblis – Color Blindness Simulator (https://www.color-blindness.com/coblis-color-blindness-simulator/) and revise the colour schemes accordingly.

**Reply to Remarks: We thank you for these remarks which have been considered in our manuscript.**

**1. Sections "Correspondence to:" and "Code/Data availability:" are now included.**

**2. Colour schemes used in maps and charts have been revised.**

**3. We have used the ACP template as suggested.**